# Comparing molnupiravir and nirmatrelvir/ritonavir efficacy and the effects on SARS-CoV-2 transmission in animal models

Robert M. Cox [1,5], Carolin M. Lieber[1,5], Josef D. Wolf [1], Amirhossein Karimi[1], Nicole A. P. Lieberman [2], Zachary M. Sticher [3], Pavitra Roychoudhury [2], Meghan K. Andrews [3], Rebecca E. Krueger [3], Michael G. Natchus[3], George R. Painter[3,4], Alexander A. Kolykhalov[3], Alexander L. Greninger[2] & Richard K. Plemper [1] ✉

Therapeutic options against SARS-CoV-2 are underutilized. Two oral drugs, molnupiravir and paxlovid (nirmatrelvir/ritonavir), have received emergency use authorization. Initial trials suggested greater efficacy of paxlovid, but recent studies indicated comparable potency in older adults. Here, we compare both drugs in two animal models; the Roborovski dwarf hamster model for severe COVID-19-like lung infection and the ferret SARS-CoV-2 transmission model. Dwarf hamsters treated with either drug survive VOC omicron infection with equivalent lung titer reduction. Viral RNA copies in the upper respiratory tract of female ferrets receiving 1.25 mg/kg molnupiravir twice-daily are not significantly reduced, but infectious titers are lowered by >2 log orders and direct-contact transmission is stopped. Female ferrets dosed with 20 or 100 mg/kg nirmatrelvir/ritonavir twice-daily show 1–2 log order reduction of viral RNA copies and infectious titers, which correlates with low nirmatrelvir exposure in nasal turbinates. Virus replication resurges towards nirmatrelvir/ritonavir treatment end and virus transmits efficiently (20 mg/kg group) or partially (100 mg/kg group). Prophylactic treatment with 20 mg/kg nirmatrelvir/ritonavir does not prevent spread from infected ferrets, but prophylactic 5 mg/kg molnupiravir or 100 mg/kg nirmatrelvir/ritonavir block productive transmission. These data confirm reports of similar efficacy in older adults and inform on possible epidemiologic benefit of antiviral treatment.

Vaccines and antivirals have helped to limit disease severity in the coronavirus disease 2019 (COVID-19) pandemic. However, extensive viral spread continues, affecting even those with vaccine or naturally-acquired immunity[1–3]. The rise of new SARS-CoV-2 variants of concern (VOC) capable of escaping preexisting immunity has undermined the hope of rapidly ending the pandemic through large-scale vaccination campaigns[1,4,5], and indirect immune imprinting may never allow vaccine boosters to be successful against future viral evolution. In 2021,

[1]Center for Translational Antiviral Research, Georgia State University Institute for Biomedical Sciences, Atlanta, GA 30303, USA. [2]Virology Division, Department of Laboratory Medicine and Pathology, University of Washington, Seattle, WA 98185, USA. [3]Emory Institute for Drug Development, Emory University, Atlanta, GA, USA. [4]Department of Pharmacology, Emory University, Atlanta, GA 30322, USA. [5]These authors contributed equally: Robert M. Cox, Carolin M. Lieber. ✉e-mail: rplemper@gsu.edu

the FDA granted emergency use authorizations for two orally available antivirals, molnupiravir and paxlovid[6,7]. Molnupiravir, a prodrug of the broad-spectrum nucleoside analog N[4]-hydroxycytidine[8], targets the viral RNA polymerase, triggering lethal viral mutagenesis[9,10]. Paxlovid consists of two components, the peptidomimetic viral main protease (M[pro]) inhibitor nirmatrelvir and the cytochrome P450 enzyme inhibitor ritonavir[11], which is required to improve pharmacokinetic properties of nirmatrelvir. Having formed covalent dead-end complexes with nirmatrelvir, M[pro] is unable to process the viral polyprotein, interrupting the viral replication cycle. Initial phase II/III clinical trials suggested superior efficacy of paxlovid compared to molnupiravir, reporting 89% versus 30% reduction in patient hospitalization and death[12,13].

These initial data and extensive public discussion of possible carcinogenic potential of molnupiravir triggered elevation of paxlovid to standard-of-care (SOC)-like status for treatment of vulnerable patient populations in the United States[14]. However, a carcinogenicity study in the Tg.rasH2 mouse model to assess carcinogenic potential of molnupiravir demonstrated that continuous dosing for 6 months was not carcinogenic[15], addressing speculations about drug safety. Recent large-scale clinical trials and retrospective assessments of therapeutic benefit have revealed that efficacy of paxlovid is substantially lower than initially seen against VOC delta[12,16,17], whereas therapeutic benefit of molnupiravir in older adults was underestimated[15,17] and molnupiravir treatment reduced the time to first recovery of high-risk vaccinated adults by ~4 days[18]. In these studies, neither drug provided significant therapeutic benefit to younger adults below 65 years of age[12,19].

Assessing the effect of post-exposure prophylactic treatment of household contacts of a confirmed SARS-CoV-2 case with paxlovid, the Evaluation of Protease Inhibition for COVID-19 in Post-Exposure Prophylaxis (EPIC-PEP) trial showed a slight reduction of preventing transmission, but benefit was not statistically significant[20,21]. Preliminary results of the comparable prevention of household transmission MOVe-AHEAD trial with molnupiravir revealed a similarly reduced likelihood of contacts to turn SARS-CoV-2 PCR test positive post-baseline, which was also not statistically significant[22]. Both prophylactic trials monitored viral RNA copy numbers as a primary endpoint. Clinical reports demonstrated rebounding virus replication in patients who have received paxlovid[2,23,24] or molnupiravir[25] and anecdotal cases of SARS-CoV-2 transmission at rebound have further questioned whether treatment may provide epidemiologic benefit through reducing the risk of SARS-CoV-2 spread. Mustelids such as mink and ferret are highly permissive for SARS-CoV-2, experiencing high virus loads predominantly in the upper respiratory tract and supporting efficient virus spread[26–28]. Using the ferret SARS-CoV-2 transmission model, we have established in previous work that oral molnupiravir blocks productive, direct-contact transmission of

infectious particles to untreated sentinel ferrets within 18 h of treatment start[27,29], but paxlovid has not been tested in the transmission model.

In this work, we compare efficacy of molnupiravir and nirmatrelvir/ritonavir in the Roborovski dwarf hamster model of severe COVID-19-like viral pneumonia and the ferret transmission model. Results demonstrate that both drugs provide significant therapeutic benefit against SARS-CoV-2 pneumonia when administered early after infection. Both the rodent and non-rodent animal model reveal a non-linear correlation of viral RNA copy number and infectious viral titer reduction after treatment with molnupiravir, whereas reductions by nirmatrelvir/ritonavir are directly proportional. Infectious virus load in the upper respiratory tract emerges as a correlate for SARS-CoV-2 transmission success in ferrets, and molnupiravir treatment completely suppresses transmission at all dose levels tested, including those not associated with a significant reduction of viral RNA copy numbers in the upper respiratory tract.

## Results

To validate sourced nirmatrelvir stocks (Fig. 1a), we determined antiviral potency on cultured VeroE6-TMPRSS2 cells against SARS-CoV-2 isolates representing lineage A (USA-WA1/2020), lineage B.1.1.7 (VOC alpha; hCoV-19/USA/CA/UCSD_5574/2020), lineage B.1.351 (VOC beta; hCoV-19/South Africa/KRISP-K005325/2020), lineage P.1 (VOC gamma; hCoV-19/Japan/TY7-503/2021), lineage B.1.617.2 (VOC delta; clinical isolate #233067), lineage B.1.1.529 (VOC omicron; hCoV-19/USA/WA-UW-21120120771/2021) lineage BA.2 (clinical isolate 22012361822A), lineage BA.2.12.1 (hCoV-19/USA/WA-CDC-UW22050170242/2022), and linage BA.4 (hCoV-19/USA/WA-CDC-UW22051283052/2022) (Fig. 1b).

### In vitro efficacy of nirmatrelvir against current VOC

Infected cells were incubated with serial dilutions of nirmatrelvir in media containing 2 μM CP-100356, a P-glycoprotein inhibitor added to limit rapid compound efflux[30]. Nirmatrelvir exhibited nanomolar potency against all VOCs tested, returning half-maximal ($EC_{50}$) and 90% maximal ($EC_{90}$) concentrations ranging from 21.1 to 327.6 nM and 32.8 to 421.4 nM, respectively, which recapitulated results of previous in vitro dose-response studies[31,32].

### Prevention of severe SARS-CoV-2 disease in Roborovski dwarf hamsters

We determined single-oral dose pharmacokinetics (PK) properties of nirmatrelvir delivered alone or in combination with ritonavir. Hamsters were administered 250 mg/kg nirmatrelvir with or without 83.3 mg/kg ritonavir. In the absence of ritonavir, 250 mg/kg nirmatrelvir delivered a plasma exposure level of 22,400 h × ng/ml (Fig. 2a, Table 1). When combined with ritonavir, nirmatrelvir plasma exposure

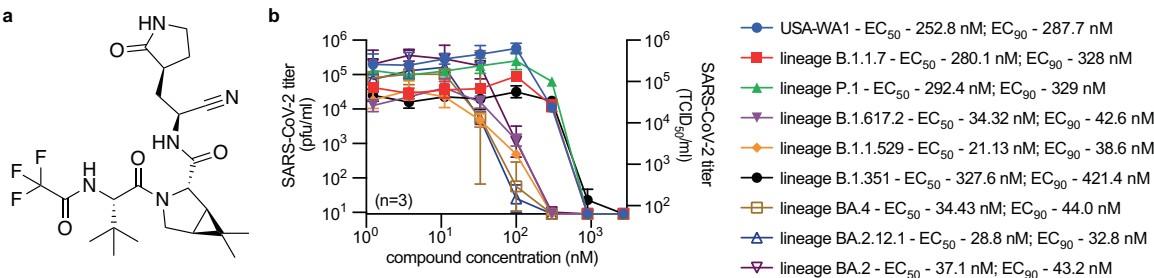

**Fig. 1 | In vitro antiviral activity of nirmatrelvir. a** Structure of nirmatrelvir. **b** Nirmatrelvir dose-response assays against SARS-CoV-2 WA1, and lineages B.1.1.7 (VOC α), B.1.351 (VOC β), P.1 (VOC γ), B.1.617.2 (VOC δ), B.1.1.529, BA.2, B1.2.12.1, and BA.4 on VeroE6-TMPRSS2 cells. The number of independent repeats (*n*) used in each experiment is shown (*n* = numbers of biologically independent samples).

Symbols represent, and lines intersect, group geometric means ± SD; numbers denote 50 and 90% inhibitory concentrations ($EC_{50}$ and $EC_{90}$, respectively), determined through non-linear 4-parameter variable slope regression modeling. Source data are provided as a Source data file.

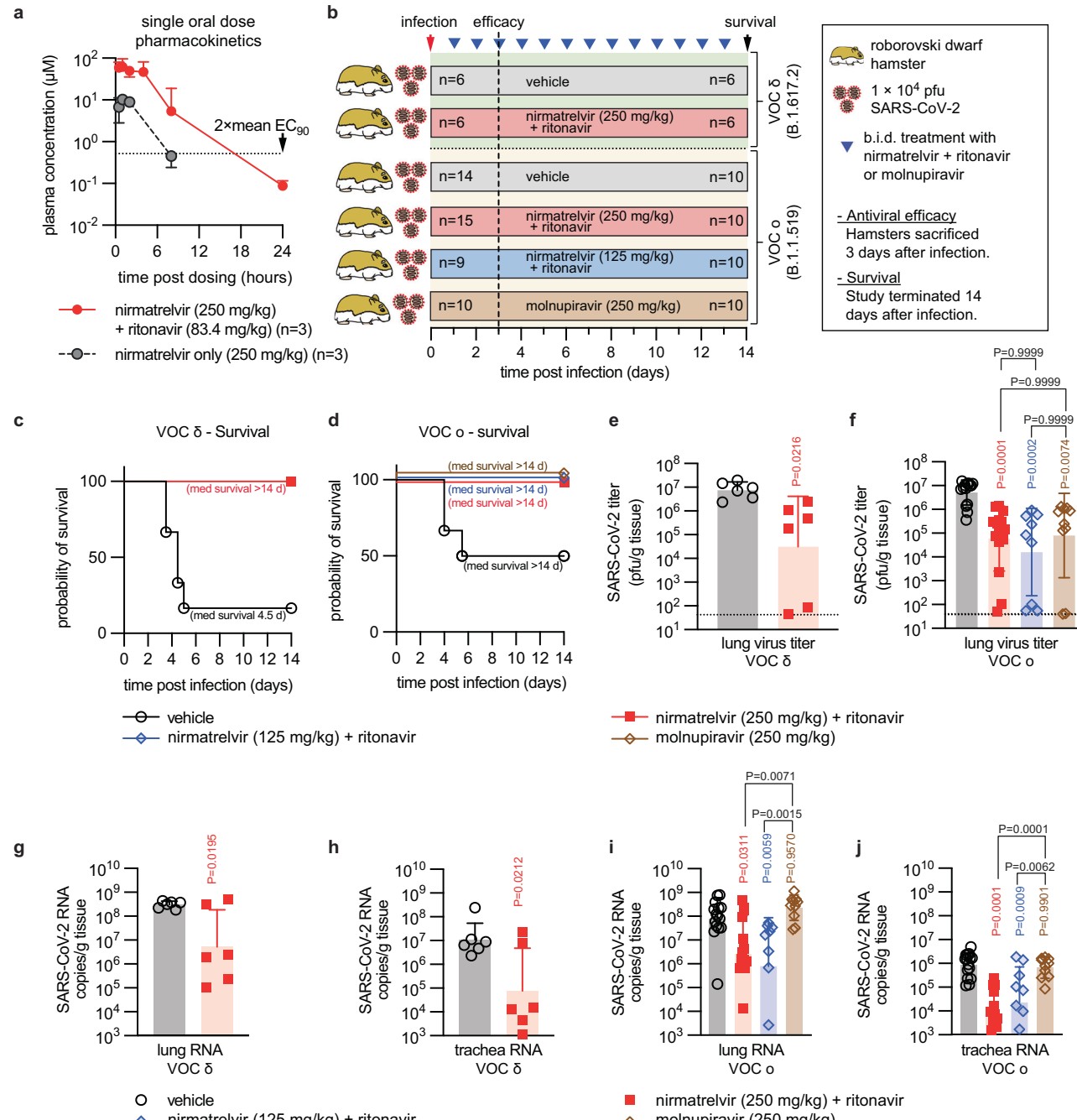

**Fig. 2 | Efficacy of paxlovid and molnupiravir in Roborovski dwarf hamsters.**
**a** Plasma concentration of nirmatrelvir over 24-h after dosing. Single oral dose of nirmatrelvir (250 mg/kg), with (red circles) or without (black circles) ritonavir (83.4 mg/kg). Symbols show group geometric means ± SD and lines intersect means. Dotted line indicates 2 × mean $EC_{90}$ against different VOC, determined in cell culture in Fig. 1b. **b** Antiviral efficacy study schematic. Animals were infected intranasally with $1 \times 10^4$ pfu of VOC delta or omicron and monitored for 14 days after infection. Clinical signs were assessed once daily (blue triangles). Viral load was determined in a duplicate set of animals 3 days after infection. The numbers of animals (*n* = number of independent animals) used in the efficacy (left) and survival

(right) studies are shown. **c, d** Survival curves of VOC delta (**c**) or omicron (**d**) infected dwarf hamsters from (**b**) Log-rank (Mantel-Cox) test, median survival (med survival) is shown. **e, f** Lung virus titers 3 days after infection with VOC delta (**e**) or omicron (**f**) as shown in (**b**); dotted lines denote limit of detection. **g–j** Lung (**g**, **i**) and trachea (**h**, **j**) viral RNA copies after infection with VOC delta (**e**) or omicron (**f**) as shown in (**b**). Symbols represent independent biological repeats; columns show group geometric means ± SD. Significance was determined using unpaired 2-sided t tests (**e**, **g**, **h**) or one-way ANOVA Tukey's post hoc test (**i**, **j**) or Kruskal–Wallis test and Dunn's post hoc test (**f**); *P* values are shown. Source data are provided as a Source data file.

was 215,000 h × ng/ml, confirming that ritonavir, equivalent to its effect in human patients, efficiently slows nirmatrelvir metabolism in the dwarf hamsters. Previously, we have demonstrated that 250 mg/kg molnupiravir yields plasma exposure of $N^4$-hydroxycytidine, the systemic metabolite of molnupiravir[33], of 376 h × ng/ml in dwarf hamsters[29]. Human patients taking paxlovid or molnupiravir show

nirmatrelvir exposure of 28,220 h × ng/ml[34,35] and $N^4$-hydroxycytidine exposure of 8200 h × ng/ml[36], respectively.

Matching the design of our previous molnupiravir dwarf hamster efficacy studies[29], we initiated oral treatment with paxlovid 12 h after intranasal infection of the dwarf hamsters with $1 \times 10^4$ pfu of VOC delta or omicron (lineage B.1.1.529), and continued treatment in a twice-daily

**Table 1 | Single dose pharmacokinetics of nirmatrelvir or nirmatrelvir/ritonavir in Roborovski dwarf hamster and human plasma**

| Drug | Species | Dose (mg/kg) | $T_{max}$ (h) | $C_{max}$ (ng/ml) | $AUC_{inf}$ (hours × ng/ml) | $t_{1/2}$ (h) | Reference |
|---|---|---|---|---|---|---|---|
| Nirmatrelvir/ ritonavir | Roborovski dwarf hamster | 250/83 | 1 | 34,200 | 215,000 | 2.21 | This study |
| Nirmatrelvir | Roborovski dwarf hamster | 250 | 1 | 5320 | 22,400 | 1.36 | This study |
| Nirmatrelvir/ ritonavir | Human | 250/100[a] | 2.75 | 2882 | 28,220 | 6.94 | 34 |
| Molnupiravir | Human | 800[a] | 1.75 | 2770 | 8200 | 1.18 | 36 |
| Molnupiravir | Roborovski dwarf hamster | 250 | 0.5 | 251 | 376 | 1.62 | 29 |

[a]Human doses are expressed as absolute amount, not mg/kg.

(b.i.d.) regimen (Fig. 2b). In the VOC omicron infection group, we included molnupiravir, dosed at 250 mg/kg b.i.d., and low-dose paxlovid (125 mg/kg nirmatrelvir and 83.3 mg/kg ritonavir) arms for comparison of effect size and/or dose-dependency of potency. The 12-h time-to-treatment-onset was chosen due to rapid disease progression in the dwarf hamster model[29], which mimics human patients at transition stage to complicated COVID-19. As we have reported for molnupiravir treatment after VOC delta infection[29], all treated dwarf hamsters met the primary efficacy endpoint and survived infection independent of VOC used, meeting the primary efficacy endpoint (Fig. 2c, d). By contrast, 83% and 50% of vehicle-treated animals had succumbed to the disease 6 days after infection with VOC delta and omicron, respectively. Drug-treated animals displayed minimal, if any, clinical signs, whereas dwarf hamsters in vehicle groups experienced hypothermia and body weight loss (Supplementary Fig. 1).

Virus burden in lungs of duplicate sets of equally infected and treated dwarf hamsters was determined 3 days after infection to determine effect size of treatment. Both VOC delta and omicron replicated efficiently in the dwarf hamsters, each reaching peak lung virus loads of ~$10^7$ pfu/g lung tissue (Fig. 2e, f). Treatment of VOC delta-infected dwarf hamsters with paxlovid significantly lowered virus burden, resembling the reduction in VOC delta lung titer that we had previously observed after treatment with molnupiravir[29]. However, effect size of treatment varied by several orders of magnitude between individual animals. We had noticed variation in effect size in molnupiravir-treated dwarf hamsters infected with VOC omicron before, but not VOC delta[29]. Head-to-head comparison of paxlovid and molnupiravir under identical experimental conditions in dwarf hamsters infected with VOC omicron demonstrated statistically significant reduction in lung virus load by either drug compared to vehicle-treated hamsters, but no statistically significant differences in effect size between both drugs or between the paxlovid dose groups (Fig. 2f).

Assessment of viral RNA copies in the lung and, in addition, trachea samples after infection with VOC delta (Fig. 2g, h) and omicron (Fig. 2i–k) revealed proportional reduction of viral RNA and infectious particles in lung samples of paxlovid-treated animals. However, no statistically significant reduction in lung viral RNA copies was detected in animals of the molnupiravir group, despite the highly significant reduction in infectious virus titer in these samples. Viral RNA copies in trachea samples of the respective paxlovid and molnupiravir groups mirrored the lung data. These results reveal similar efficacy of paxlovid and molnupiravir in reducing infectious virus titers in lung and preventing lethal disease in the dwarf hamster model. The non-linear correlation between reduction of infectious viral titers and viral RNA copy numbers in molnupiravir recipients suggested that a large proportion of viral RNA synthesized in molnupiravir-experienced animals may be bio-inactive.

To assess possible virus rebound after treatment, we discontinued treatment with either paxlovid (250 mg/kg nirmatrelvir/ritonavir, b.i.d.) or molnupiravir (250 mg/kg b.i.d.) five days after infection with VOC omicron and determined lung virus load on days 5 and 9 (Fig. 3a). All treated animals survived without developing clinical signs, whereas vehicle-treated animals experienced severe weight loss and hypothermia, and ~50% succumbed to infection by day 5 (Fig. 3b, c).

Infectious VOC omicron titers in lungs were reduced equally by paxlovid and molnupiravir compared to peak titers (day 3) in vehicle-treated animals (Fig. 3d). On day 5, no statistically significant differences between either treatment group and vehicle-treated animals were detectable. Lung virus load in all groups approached the limit of detection by day 9 without signs of virus rebound in the treatment groups.

## Paxlovid treatment of SARS-CoV-2 infection in ferrets

To assess the impact of nirmatrelvir treatment on upper respiratory disease, we employed the SARS-CoV-2 ferret infection model[27,28,37,38]. To ensure cross-species consistency of nirmatrelvir PK, we again administered the compound first in a single oral dose at two levels, 20 mg/kg and 100 mg/kg, each in combination with 6 mg/kg ritonavir. Overall nirmatrelvir plasma exposure in ferrets was 32,700 h × ng/ml and 49,500 h × ng/ml for the 20 and 100 mg/kg groups, respectively (Fig. 4a; Table 2), indicating a dose-dependent, but not dose-proportional, PK profile of paxlovid.

Both dose levels were advanced to efficacy testing after intranasal infection of ferrets with $1 × 10^5$ pfu of SARS-CoV-2 isolate USA-WA1/2020 (lineage A), which most efficiently replicates and spreads in ferrets[27,29,37,39,40] and other members of the Mustela genus[41,42]. Treatment was initiated 12 h after infection and continued b.i.d., and nasal lavages collected twice daily to determine shed virus load (Fig. 4b). Characteristic for this model[27,28,37,38], SARS-CoV-2-infected ferrets displayed no overt clinical signs (Supplementary Fig. 2). Virus replicated efficiently in vehicle-treated animals, reaching plateau titers of ~$10^5$ pfu/ml shed virus and $10^5$ RNA copies/μl nasal lavage 48 h after infection, which were maintained until study day 4 (Fig. 4c, d). By study end, both infectious viral titers and viral RNA copies in lavages of the 20 mg/kg nirmatrelvir/ritonavir group were reduced by ~1.4 log orders.

Ferrets receiving 100 mg/kg nirmatrelvir/ritonavir experienced a significant titer reduction in nasal lavages, but did not fully clear the infection. Rather, we noted a slight increase of virus replication in both nirmatrelvir/ritonavir dose groups 3.5 days after treatment start. Whole genome sequencing of virus populations in nasal lavage and turbinate samples taken from these animals revealed that most had acquired an N501T or Y453F substitution in the spike protein, which are characteristic for SARS-CoV-2 adaptation to replication in mustelids[27] (Supplementary Fig. 3), but no signature resistance mutation to nirmatrelvir in nsp5[43–45] emerged as allele-dominant (uploaded as NCBI BioProject PRJNA894555). At study end, infectious titers and viral RNA copies in nasal turbinate tissues were statistically significantly reduced in nirmatrelvir/ritonavir-treated animals compared to the vehicle group (Fig. 4e, f).

## Effect of paxlovid treatment on transmission from SARS-CoV-2-infected ferrets

To examine the effect of nirmatrelvir/ritonavir treatment on SARS-CoV-2 spread, we subjected treated ferrets to a direct-contact transmission study, first co-housing infected and treated source ferrets with untreated contact animals 42 h after treatment initiation (Fig. 5a). Source animals were infected intranasally with SARS-CoV-2 WA1 as before, followed by nirmatrelvir/ritonavir treatment in a b.i.d. regimen

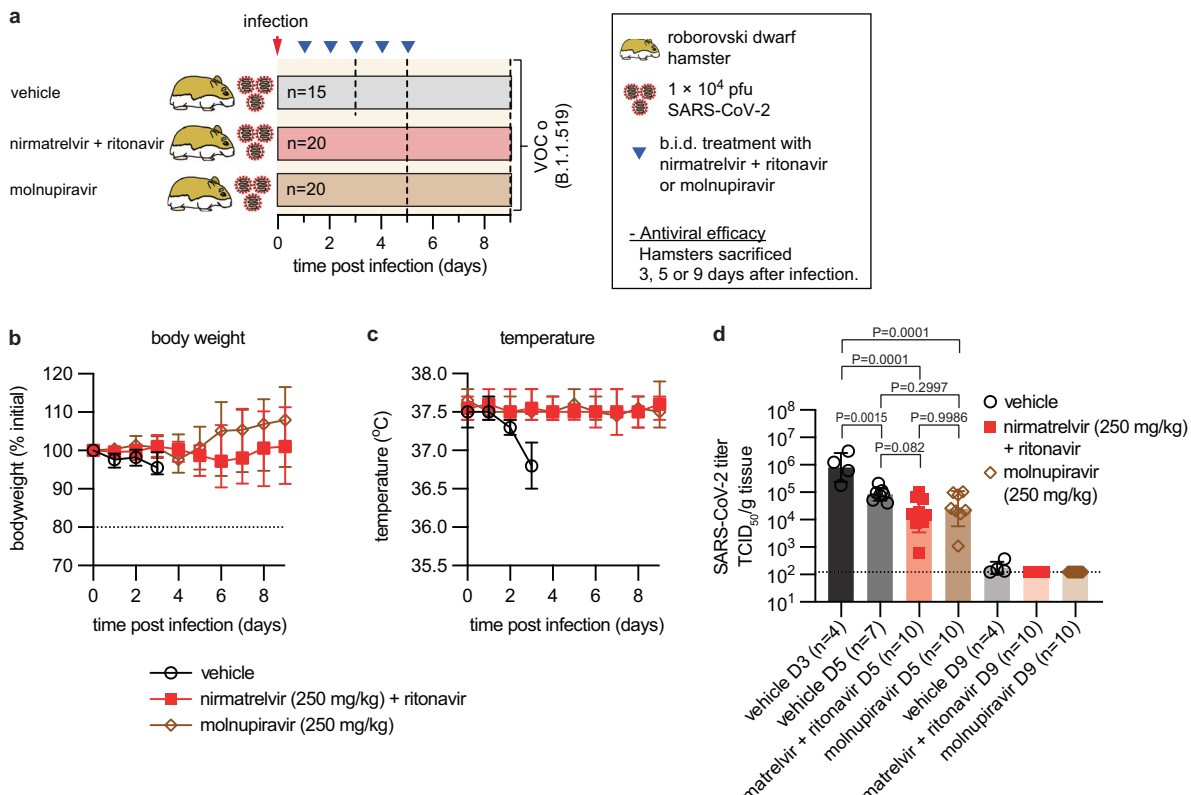

**Fig. 3 | VOC omicron lung virus load following antiviral treatment. a** Schematic of the study design. Treatment with nirmatrelvir/ritonavir or molnupiravir was initiated 12 h after infection, continued b.i.d., and terminated 5 days after infection. **b**, **c** Roborovski dwarf hamster body weight (**b**) and temperature (**c**) in animals infected with VOC omicron and treated with drug or vehicle. Symbols show, and lines intersect, group means ± SD. Dotted line in (**b**) denotes predefined endpoint. **d** Infectious virus titers in lung tissue extracted on days 3 (D3; vehicle only), 5 (D5), and 9 (D9) after infection. The number of animals ($n$ = independent animals) used in each experiment is specified. Symbols represent independent biological repeats (individual animals); columns denote geometric means ± SD. Two-way ANOVA with main effects only, Tukey's multiple comparison post hoc test; $P$ values are shown. Dotted line denotes limit of detection. Source data are provided as a Source data file.

at 20 or 100 mg/kg. For comparison with the only drug in clinical use that reportedly suppresses viral transmission in this model[27,29], we added molnupiravir treatment groups, dosed orally b.i.d. at 1.25 mg/kg, 2.5 mg/kg, or 5 mg/kg. All treatments were initiated 12 h after infection. Equivalent amounts of shed infectious viral particles were detected in all source animal groups at the time of treatment start, confirming productive infection of all animals (Fig. 5b, c, Supplementary Fig. 4). None of the animals developed clinical signs (Supplementary Fig. 5). Shed virus and RNA copy numbers of vehicle and nirmatrelvir/ritonavir-treated sources resembled that of the initial efficacy study (Supplementary Fig. 6), whereas shed infectious titers in two of the three molnupiravir arms rapidly declined, reaching detection level ~3 days after treatment start (Fig. 5b–d). Recapitulating the dwarf hamster results, viral RNA copy numbers and infectious viral titers in lavages of molnupiravir-treated ferrets followed a non-linear correlation, amounting only to an ~0.5 log order difference to vehicle-treated animals in the 1.25 mg/kg molnupiravir group by study end (Fig. 5e), but a statistically highly significant reduction in infectious titers of approximately two orders of magnitude (Fig. 5c).

SARS-CoV-2 spread rapidly to sentinels of vehicle-treated source animals. As noted before[27,29], molnupiravir treatment of source animals rapidly reduced infectious viral titers, suppressing all transmission to untreated naïve contacts in all molnupiravir dose groups including the 1.25 mg/kg group (Fig. 5c, e). By contrast, viral spread to sentinels of ferrets receiving 20 mg/kg b.i.d. nirmatrelvir/ritonavir was delayed, but ultimately all contacts were infected and viral titers in nasal lavages were virtually identical to those of sentinels of vehicle-treated animals,

indicating uninterrupted transmission. Treating source ferrets with 100 mg/kg nirmatrelvir/ritonavir reduced transmission rates, but productive viral transfer still occurred in one of the three contact pairs examined (Fig. 5b, d). Again, virus populations in lavage and turbinate samples of these animals had acquired mustelid-characteristic species adaptation mutations in the spike protein in most cases (Supplementary Fig. 7), but whole genome sequencing did not reveal any allele-dominant resistance mutation to nirmatrelvir in nsp5 (uploaded as NCBI BioProject PRJNA894555).

Viral titers and RNA copies in nasal turbinates determined 4 days after infection in source animals and 4 days after the end of co-housing in contact animals (study day 8) revealed turbinate titers and viral RNA copies in sentinels of sources that had received 20 mg/kg nirmatrelvir/ritonavir indistinguishable from those of animals that had been co-housed with vehicle-treated sources (Fig. 5d, e). One of three contacts of the 100 mg/kg nirmatrelvir/ritonavir recipients showed vehicle-equivalent viral burden in upper respiratory tissues at study end. No infectious particles or viral RNA was detectable in nasal turbinates extracted from contacts of all molnupiravir treatment groups (Fig. 5h, i), confirmed a sterilizing effect of molnupiravir on transmission in all dose groups. Comparison of infectious titers (Fig. 5h) versus viral RNA copies (Fig. 5i) in turbinates of the different molnupiravir dose level source animals underscored the notion that a large proportion of viral RNA synthesized in the presence of molnupiravir was bio-inactive.

We hypothesized that the discrepancy between comparable performance of nirmatrelvir/ritonavir and molnupiravir in dwarf hamster lung tissue and differential effect on viral spread may be due to poor

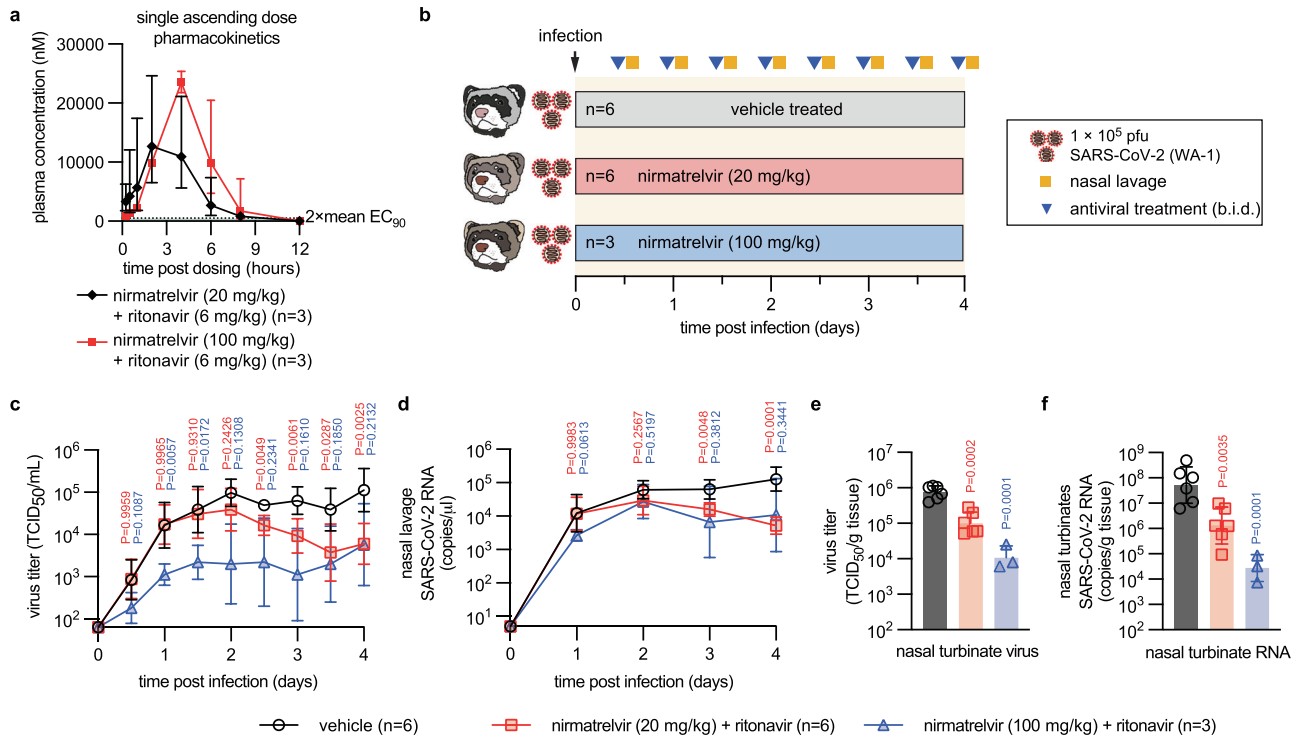

**Fig. 4 | Efficacy of paxlovid in ferrets. a** Plasma concentration of nirmatrelvir determined over a 12-h period after a single oral dose. Nirmatrelvir was dosed at 20 mg/kg (black diamonds) or 100 mg/kg (red squares), each in combination with 6 mg/kg ritonavir. Symbols represent group means ± SD and lines intersect means. Dotted line indicates 2 × mean $EC_{90}$ against different VOC, determined in cell culture in Fig. 1b. **b** Antiviral efficacy study schematic. Ferrets ($n = 15$ total) were infected intranasally with $1 \times 10^5$ pfu of SARS-CoV-2 USA/WA-1. After 12 h, groups of ferrets were treated orally with vehicle or 20 mg/kg or 100 mg/kg nirmatrelvir/ritonavir. The number of animals ($n$ = independent animals) used in each experiment is specified. **c**, **d** Infectious SARS-CoV-2 titers (**c**) and SARS-CoV-2 RNA copies (**d**) in nasal lavages of animals shown in (**b**). **e**, **f** Infectious titers (**e**) and SARS-CoV-2 RNA copies (**f**) in nasal turbinates extracted four days after infection. Symbols in (**c**–**f**) represent independent biological repeats (virus load of individual animals); lines (**c**, **d**) intersect, and columns (**e**, **f**) show, group geometric means ± SD. Statistical analysis with one-way (**e**, **f**) or two-way (**c**, **d**) ANOVA with Dunnett's (**c**–**f**) post hoc tests; $P$ values are shown. Source data are provided as a Source data file.

**Table 2 | Single dose pharmacokinetics of nirmatrelvir/ritonavir in ferret and human plasma**

| Drug | Species | Dose (mg/kg) | $T_{max}$ (h) | $C_{max}$ (ng/ml) | $AUC_{inf}$ (hours × ng/ml) | $t_{1/2}$ (h) | Reference |
|---|---|---|---|---|---|---|---|
| Nirmatrelvir/ritonavir | Ferret | 20/6 | 2.33 | 8990 | 32,700 | 1.05 | This study |
| Nirmatrelvir/ritonavir | Ferret | 100/6 | 4 | 11,800 | 49,500 | 0.74 | This study |
| Nirmatrelvir/ritonavir | Human | 250/100[a] | 2.75 | 2882 | 28,220 | 6.94 | 34 |
| Molnupiravir | Human | 800[a] | 1.75 | 2770 | 8200 | 1.18 | 36 |
| Molnupiravir | Ferret | 4 | 1.7 | 907 | 3421 | 8.2 | 33 |
| Molnupiravir | Ferret | 7 | 1.5 | 2851 | 7569 | 9.2 | 33 |
| Molnupiravir | Ferret | 20 | 1.8 | 3992 | 18,793 | 4.7 | 33 |
| Molnupiravir | Macaque | 130 | 1.62 | 2644 | 25,299 | 1.77 | 33 |

[a]Human doses are expressed as absolute amount, not mg/kg.

nirmatrelvir exposure in upper respiratory tract tissues. Assessing drug levels at anticipated peak (3 h after dosing, ~30 min after plasma $C_{max}$) and trough (12 h after dosing) revealed statistically significantly lower nirmatrelvir levels in nasal turbinates compared to all other soft tissues but brain (Fig. 5j).

**Prophylactic treatment of uninfected direct contact ferrets**
In the EPIC-PEP and MOVe-AHEAD trials, prophylactic administration of paxlovid or molnupiravir, respectively, to adults exposed to household members did not significantly reduce the risk of infection[20–22]. We prophylactically treated uninfected ferrets with 20 mg/kg or 100 mg/kg nirmatrelvir/ritonavir or 5 mg/kg molnupiravir, all in b.i.d. regimen, followed by co-housing with infected, but untreated, source animals 12 h later (Fig. 6a). Treatment of the

sentinels was continued b.i.d. for 5.5 days, shed virus in nasal lavages of source and sentinels monitored daily, and virus load in the upper respiratory tract determined 6 days after infection.

We detected shed virus in lavages of all contacts receiving vehicle within 24 h of co-housing, indicating productive transmission (Fig. 6b). Again, none of the animals developed clinical signs (Supplementary Fig. 8). Prophylactic 20 mg/kg nirmatrelvir/ritonavir delayed transmission by ~48 h, but ultimately all sentinels were productively infected and shed virus titers reached by study end were indistinguishable from those of the vehicle group. In contrast, no infectious SARS-CoV-2 particles emerged in lavages of sentinels prophylactically treated with 100 mg/kg nirmatrelvir/ritonavir or 5 mg/kg molnupiravir. Quantitation of viral RNA in lavage samples groups largely mirrored the virus titer profiles (Fig. 6c). However, low amounts (<$10^3$ RNA copies/µl) of

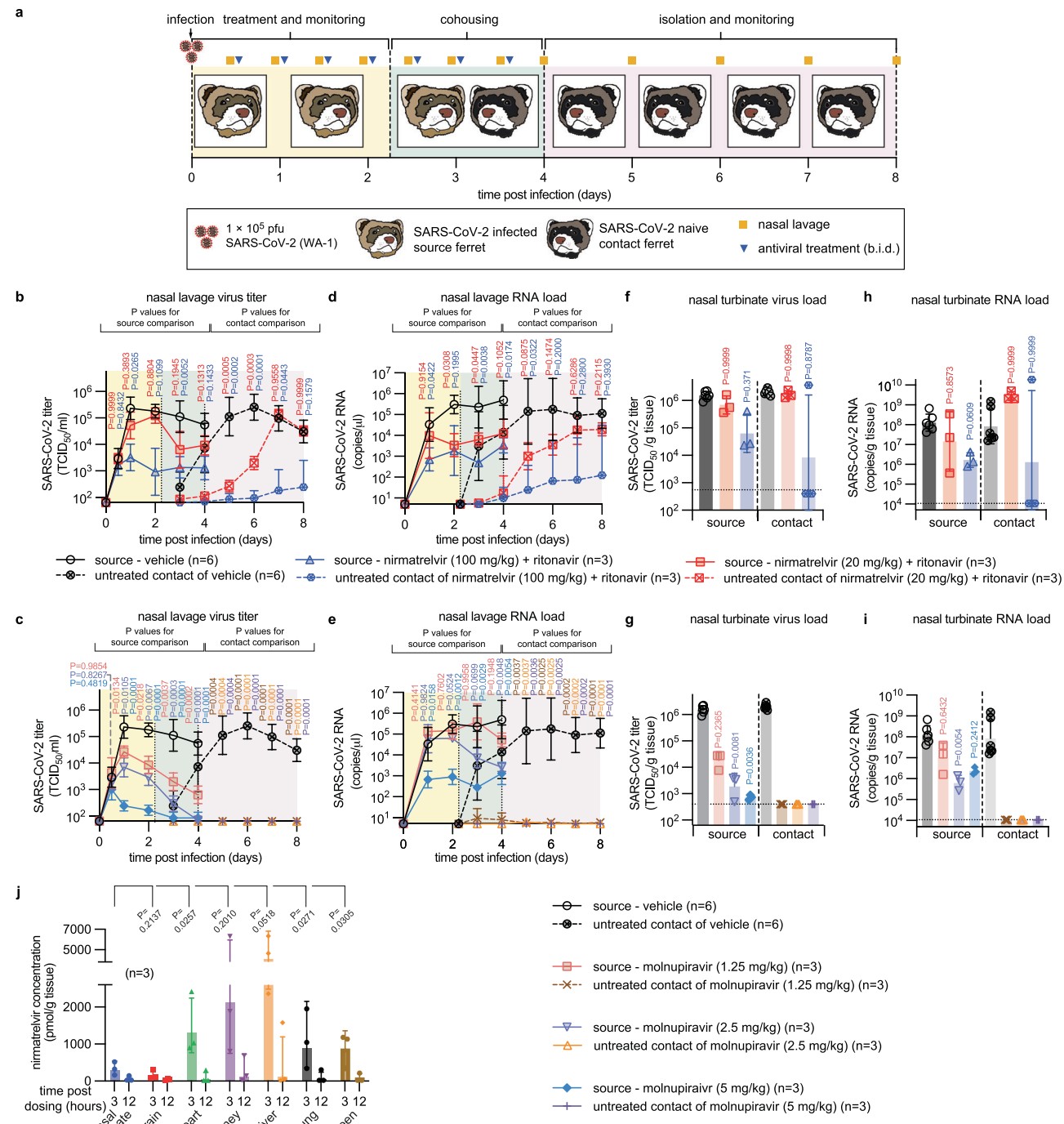

**Fig. 5 | Effect of treatment with paxlovid or molnupiravir on SARS-CoV-2 transmission. a** Transmission study schematic. **b–i** Direct-contact transmission study. Results are disaggregated into nirmatrelvir/ritonavir (**b–e**) and molnupiravir (**f–i**) treatment groups; vehicle-treated animals are shown in both sets. Infectious titers (**b, c**) and RNA copies (**d, e**) of SARS-CoV-2 present in nasal lavages of source and contact ferrets. Nasal turbinate titers (**f, g**) and RNA copies (**h, i**) in nasal turbinates extracted from source and contact animals on study days 4 and 8, respectively. Symbols (**b–e**) represent, and lines intersect, geometric means ± SD; x-axes denote level of detection. Symbols (**f–i**) represent independent biological repeats (results from individual animals); columns denote geometric means ± SD; dotted lines specify level of detection. **j** nirmatrelvir exposure in selected ferret organs at anticipated peak and trough after single-dose administration of nirmatrelvir + ritonavir. The number of animals (*n* = independent animals) used in each experiment is specified. Symbols represent independent biological repeats (results from individual animals); columns denote means ± SD. Statistical analysis with one-way (**f–i**) or two-way (**b–e**) ANOVA with Dunnett's post hoc tests (**b–e**), Kruskal–Wallis test with Dunn's post hoc tests (**f–i**), or ratio paired two-sided t test (**j**); *P* values are shown. Source data are provided as a Source data file.

SARS-CoV-2 RNA were present also in lavages of the 5 mg/kg molnupiravir and 100 mg/kg nirmatrelvir/ritonavir groups, demonstrating that co-housing with untreated source animals efficiently exposed sentinels of all four study arms to SARS-CoV-2. Infectious virus load in nasal turbinates of animals treated with 20 mg/kg nirmatrelvir/ritonavir was statistically indistinguishable from that found in the vehicle group (Fig. 6d). Turbinates of one of the contacts treated with 100 mg/kg nirmatrelvir/ritonavir harbored some infectious particles, whereas turbinates of all sentinels receiving 5 mg/kg molnupiravir did not contain infectious SARS-CoV-2 at study end. We detected high RNA

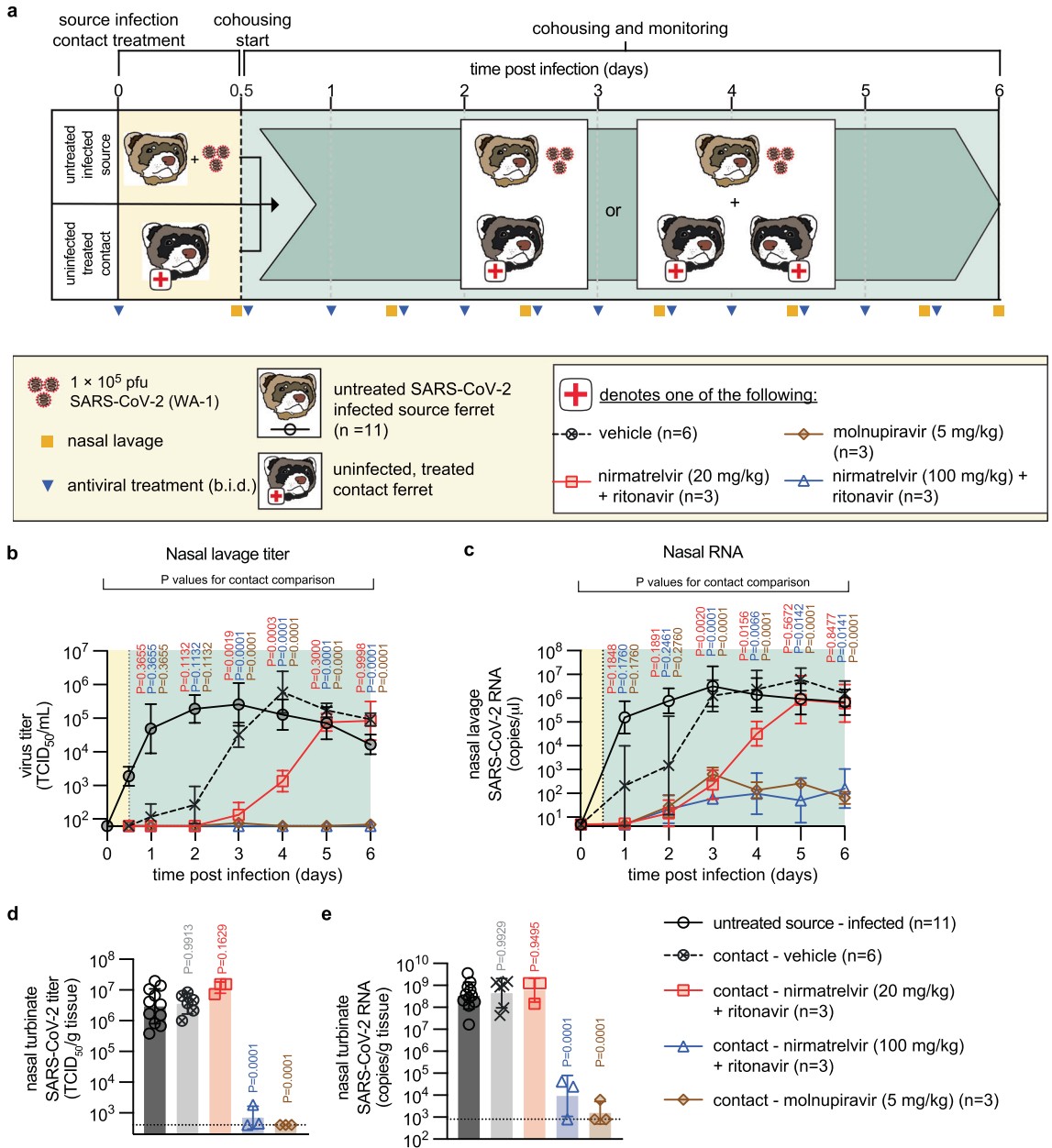

**Fig. 6 | Prophylactic treatment of close contacts with paxlovid or molnupiravir. a** Prophylaxis study schematic. **b, c** Infectious titers (**b**) and RNA copies (**c**) of SARS-CoV-2 present in nasal lavages of source and contact ferrets. The number of animals (*n* = independent animals) used in each experiment is specified. **d, e** Nasal turbinate titers (**d**) and RNA copies (**e**) in nasal turbinates extracted from source and contact animals on study day 6. Symbols (**b, c**) represent, and lines intersect, geometric means ± SD; x-axes denote level of detection. Symbols (**d, e**) represent independent biological repeats (results from individual animals); columns denote geometric means ± SD; dotted lines specify level of detection. Statistical analysis with one-way (**d, e**) or two-way (**b, c**) ANOVA with Dunnett's (**b–e**) post hoc tests; *P* values are shown. Source data are provided as a Source data file.

copy numbers in all 20 mg/kg nirmatrelvir/ritonavir recipients, which were statistically equivalent to those derived from vehicle-treated animals (Fig. 6e). Low levels of viral RNA were furthermore observed in the turbinates of two animals of the 100 mg/kg nirmatrelvir/ritonavir group and one animal in the 5 mg/kg molnupiravir group. One 100 mg/kg nirmatrelvir/ritonavir animal and the other two 5 mg/kg molnupiravir recipients remained viral RNA free.

## Discussion

The retrospective analyses of members of Clalit Health Services[16,19] and a retrospective cohort study from the Hong Kong VOC omicron wave[17] demonstrated that both paxlovid[16] and molnupiravir[19] significantly lowered the risk of progression to severe disease and deaths in older

adults. However, a direct comparison of drug efficacy was not possible due to imbalances in baseline characteristics of the study groups such as patient comorbidities, age, and vaccine status[46]. To assess efficacy of both drugs against COVID-19 of different severity under equivalent experimental conditions, we employed the Roborovski dwarf hamster and ferret models of SARS-CoV-2 infection.

Establishing a correlation between animal and human dose levels is complex. Interspecies allometric scaling to convert dose-based systemic exposure from animals to humans can confound predictive power of animal models[47]. Although no scaling was required for our study since plasma exposure levels of both nirmatrelvir[34] and N[4]-hydroxycytidine[36], the systemic metabolite of molnupiravir[33] were known (included in Tables 1, 2), comparison dose selections can still be

undermined by, for instance, host species-specific differences in drug plasma binding, tissue penetration, viral replication kinetics, and drug half-life.

To place ferret dose levels in context to clinical data, we therefore evaluated for each drug individually treatment effect size in the animal model and human patients. Since accurate and reproducible quantitation of infectious viral titers in human nasopharyngeal swabs is difficult[48], viral RNA copies have become the clinical standard and RNA copy number reduction in the upper respiratory tract represents the sole quantitative biomarker that is available across multiple species including humans. Clinical trials reported an ~0.5 log order reduction of viral RNA copies by molnupiravir[49], and 0.89–0.93 log orders by paxlovid[12,21]. An equivalent effect size was achieved in the ferret upper respiratory tract by 1.25 mg/kg molnupiravir (~0.5 log order reduction) and 20 mg/kg nirmatrelvir/ritonavir (1.4 log order reduction), each administered in b.i.d. regimen.

In the case of paxlovid, the correlation between viral RNA copy and infectious titer reductions was approximately linear in the models. A viral mutagen such as molnupiravir, however, once incorporated into viral RNA, can destroy bioactivity of this RNA without proportionally reducing the number of physical RNA copies, which may result in underappreciation of its antiviral effect based on RNA copy numbers alone. Both the dwarf hamster and ferret model illustrate that this difference between reduction of infectious virus titer and viral RNA molecules becomes most pronounced at low molnupiravir exposure, veering in some cases from statistically insignificant (RNA copies) to several orders of magnitude (infectious titer) in the same sample.

It is undocumented at present whether a similar discrepancy between viral RNA copy number and infectious viral titer reduction applies to clinical samples. Since the effect is consistent across rodent and non-rodent host models and appears to be directly linked to the mechanism of action of molnupiravir, quantitating viral RNA copies without assessing RNA bioactivity may be at risk of being misleading in the case of viral mutagens such as molnupiravir and may, therefore, not represent an informative primary endpoint for clinical trials. Support for this notion comes from qualitative infectious viral load data determined in the phase 2a clinical trial of molnupiravir[49]. After a 5-day treatment course, infectious virus particles could be isolated from patients in the placebo group, but not from patients receiving molnupiravir, although these patients experienced only an ~0.5 log order decrease in viral RNA copies in the upper respiratory tract.

An antiviral effect size-based approach could not be applied to the dwarf hamster model of acute SARS-CoV-2 pneumonia, since viral RNA copy number reductions in the human lower respiratory tract are unknown. Based on our PK information, $N^4$-hydroxycytidine[29] plasma exposure in dwarf hamsters was over 20-fold lower than that in humans, whereas nirmatrelvir exposure exceeded that achieved in humans by over 7.5-fold, suggesting that the dwarf hamster model may overestimate efficacy of nirmatrelvir/ritonavir against acute lung disease.

Acknowledging this caveat, the dwarf hamster model supported three major conclusions: (i) both drugs met the primary efficacy endpoint, ensuring complete survival of treated animals; (ii) both drugs statistically significantly reduced lung virus burden compared to vehicle-treated animals; and (iii) treatment benefit provided by either drug showed variations between individual animals when dwarf hamsters were infected with VOC omicron. We have recently reported that in the case of molnupiravir, biological sex of the treated dwarf hamsters contributes to effect[29]. Although not powered for a full statistical analysis, nirmatrelvir/ritonavir performance in dwarf hamsters did not suggest an equivalent role of biological sex on effect size against VOC omicron. We hypothesize that differences in the kinetics of viral host invasion between individual dwarf hamsters may contribute to this variation of effect size. In toto, the dwarf hamster model recapitulated the main conclusions of retrospective clinical studies in older adults[17];

both drugs mitigated advance to severe disease, preventing acute lung injury and death.

Prolonged self-isolation periods after a positive SARS-CoV-2 test impose considerable social hardship on the individual patient and economic burden on societies[50–52], which has resulted in eroding public compliance with self-quarantine during the course of the pandemic[53]. In addition to direct therapeutic benefit, pharmacologically shortening the duration of the infectious period will improve patient quality of life and may provide epidemiologic benefit through interruption of viral transmission chains[54,55]. We have established efficient suppression of SARS-CoV-2 transmission in the ferret model by molnupiravir dosed at 5 mg/kg b.i.d.[27,29] and experimental nucleoside analog antivirals[37] in previous work, and demonstrated in this study that therapeutic administration of 1.25 mg/kg molnupiravir or 100 mg/kg nirmatrelvir/ritonavir to ferrets fully or partially, respectively, suppressed transmission to untreated direct contacts. At these dose levels, either drug reduced shed infectious viral titers to ~$10^3$ TCID$_{50}$ units/ml nasal lavage fluid, which appears to represent a threshold for productive direct-contact SARS-CoV-2 transmission in the ferret model. SARS-CoV-2 transmitted in all pairs of the 20 mg/kg nirmatrelvir/ritonavir group, which reduced viral titers to ~$10^4$ TCID$_{50}$ units/ml nasal lavage fluid, indicating that infectious viral titer in the ferret upper respiratory tract is a correlate for transmission success.

Whole genome sequencing of virus populations isolated from paxlovid-experienced ferrets provided no evidence that replication resurgence and transmission both from and to paxlovid-treated animals in our study were supported by emerging resistance to nirmatrelvir[43–45]. Rather, poor control of SARS-CoV-2 spread in ferrets coincided with significantly lower nirmatrelvir levels in the upper respiratory tract than lung and other soft tissues but brain, identifying nasal turbinates as a privileged organ of low nirmatrelvir exposure. This effect could be ferret-specific, since tissue drug exposure levels cannot be readily determined in humans. However, rebounding virus replication in paxlovid recipients[2,23,24] and the lack of efficacy in patients <65 years of age, in which SARS-CoV-2 infection resides primarily in the upper respiratory tract, are consistent with low nirmatrelvir exposure also in human upper respiratory tissues.

Prophylactic dosing of contact ferrets with 20 mg/kg paxlovid did not prevent transmission from infected animals, recapitulating the outcome of the failed EPIC-PEP trial[20,21] trial. Interestingly, 20 mg/kg nirmatrelvir/ritonavir was also the ferret dose that mediated a similar viral RNA copy number reduction to that seen in humans receiving paxlovid. Since also prophylactic molnupiravir did not prevent infection in the MOVe-AHEAD trial[22], an equally negative outcome of prophylactic 1.25 mg/kg molnupiravir was likely, prompting us not to commit additional animals for an experimental confirmation. However, 5 mg/kg prophylactic molnupiravir fully, and 100 mg/kg prophylactic nirmatrelvir/ritonavir partially, suppressed infection, indicating that higher dose levels, as long as compatible with safety margins, provide prophylactic efficacy in ferrets.

Trials with appropriately selected endpoints will be required to determine whether molnupiravir over-proportionally reduces infectious SARS-CoV-2 titers, compared to viral RNA copy numbers, also in the human upper respiratory tract. Due to differences in the dynamics of viral host invasion and replication in humans versus animals, extrapolation of therapeutic time windows from animal models to human patients is challenging. Despite the potential of host species-specific effects, this study affirms retrospective clinical analyses that early treatment of older adult patients at elevated risk of progression to severe COVID-19 with either paxlovid or molnupiravir will provide significant therapeutic benefit. We demonstrate in two animal model species, one rodent and one non-rodent, that infectious particle titers, but not viral RNA copy numbers, should be assessed to determine efficacy of a viral mutagen such as molnupiravir. Infectious viral load in the upper respiratory tract emerged as a correlate for SARS-CoV-2

transmission success in ferrets. Therapeutic molnupiravir, administered at a dose not associated with a significant reduction of viral RNA copy numbers, completely suppressed transmission in ferrets.

## Methods

### Ethics and inclusion statement

All in vivo studies were performed in compliance with the Guide for the Care and Use of Laboratory Animals, National Institutes of Health guidelines, and the Animal Welfare Act Code of Federal Regulations. Experiments with SARS-CoV-2 involving ferrets were approved by the Georgia State Institutional Animal Care and Use Committee under protocol A20031. Experiments with SARS-CoV-2 involving Roborovski dwarf hamsters were approved by the Georgia State Institutional Animal Care and Use Committee under protocol A21019. All experiments using infectious SARS-CoV-2 strains were approved by the Georgia State Institutional Biosafety Committee under protocol B20016 and performed in BSL-3/ABSL-3 facilities at the Georgia State University.

### Study design

Male and female Roborovski dwarf hamsters (*Phodopus roborovskii*, 3–4 months old) and female ferrets (*Mustela putorius furo*, 6–10 months old) were used as in vivo models to assess the therapeutic efficacy of orally administered nirmatrelvir against infection with SARS-CoV-2. Prior to studies, hamsters and ferrets were rested for at least five days. Roborovski dwarf hamsters were used to study the effects of nirmatrelvir treatment on severe disease associated with lower respiratory tract infection, acute lung injury, and death. Ferrets were used to examine the effect of nirmatrelvir on upper respiratory infection and transmission. SARS-CoV-2 infections were established in animals through intranasal inoculation. For hamsters, animals were monitored regularly for clinical signs and viral loads were determined in respiratory tract tissues at endpoint. For ferrets, upper respiratory tract viral titers were assessed at least once daily through nasal lavages. In addition, upper respiratory tract tissues were harvested at endpoint to assess upper respiratory tract viral loads. Virus loads were determined by plaque assays or $TCID_{50}$ and RT-qPCR quantitation.

### Cells

VeroE6-TMPRSS2 (BPS Bioscience #78081) and Calu-3 (ATCC HB-55™) cells were cultivated at 37 °C with 5% $CO_2$ in Dulbecco's modified Eagle's medium (DMEM) supplemented with 7.5% heat-inactivated fetal bovine serum (FBS). All cells were authenticated phenotypically and tested routinely in 3-months intervals for absence of mycoplasma contamination prior to use.

### Viruses

SARS-CoV-2 USA-WA1 (lineage A, isolate USA-WA1/2020, BEI cat# NR-52281) was obtained from BEI resources and amplified on Calu-3 cells. SARS-CoV-2 isolates for lineage B.1.1.7 (VOC α; hCoV-19/USA/CA/UCSD_5574/2020), lineage B.1.351 (VOC β; hCoV-19/South Africa/KRISP-K005325/2020), lineage P.1 (VOC γ; hCoV-19/Japan/TY7-503/2021), lineage B.1.617.2 (VOC δ; clinical isolate #233067), lineage B.1.1.529 (VOC o; hCoV-19/USA/WA-UW-21120120771/2021) lineage BA.2 (clinical isolate 22012361822A), lineage BA.2.12.1 (hCoV-19/USA/WA-CDC-UW22050170242/2022), and lineage BA.4 (hCoV-19/USA/WA-CDC-UW22051283052/2022) were obtained from UW Virology and amplified on Calu-3 cells. All viruses were titered on VeroE6-TMPRSS2 cells and virus stocks were authenticated by whole genome next-generation sequencing prior to use.

### Virus yield reduction

Nirmatrelvir was dissolved in water-free DMSO at 10 mM stock concentration and stored in single-use aliquots at −80 °C. Cells were seeded 14 h prior to experimentation in 24-well plates at $2 \times 10^5$ cells per well in 400 µl per well DMEM + 7.5% FBS supplemented with 2 µM

CP-100356. Serial dilutions were generated in culture media containing 2 µM CP-100356 spanning a final concentration range from 2700 to 1.4 nM (3-fold serial dilution; 8 steps; dilution series were prepared 2-fold overconcentrated (54000 to 2.8 nM)). Every concentration of each test article was assessed in 3 biological (independent) repeats. Vehicle-treated control cultures DMSO volume equivalents corresponding to the highest test compound concentration. Serial dilutions were transferred to target wells (500 µl per well), immediately followed by infection with SARS-CoV-2 (multiplicity of infection 0.1 pfu per cell) in 100 µl per well (viral dilution media contained 2 µM CP-100356). Infected plates were incubated in a humidified incubator at 37 °C and 5% $CO_2$ for 48 h. Progeny virus titers in culture supernatants were determined through plaque-forming or $TCID_{50}$ assay on Vero-E6-TMPRSS2 cells. Inhibitory concentrations were calculated based on 4-parameter variable slope regression models.

### SARS-CoV-2 titration by plaque assay

All samples were serially diluted in DMEM media supplemented with 2% FBS and antibiotic-antimycotic solution (Gibco). Dilutions were added to VeroE6-TMPRSS2 cells seeded 24 h earlier in 12-well plates at $3 \times 10^5$ cells per well. Dilutions were allowed to adsorb for 2 h at 37 °C. Following adsorption, inoculum was removed, and cells were overlaid with 1.2% Avicel (FMC BioPolymer) in DMEM supplemented with antibiotic-antimycotic solution and incubated at 37 °C with 5% $CO_2$. After incubating for 3 days, Avicel was removed, cells were washed with PBS and fixed with 10% neutral buffered formalin. Plaques were visualized using 1% crystal violet.

### Single-dose pharmacokinetics in hamsters

Groups of hamsters were administered a single oral dose of nirmatrelvir (250 mg/kg; 0.5% MC with 2% Tween80) with or without ritonavir (83.4 mg/kg; 20% ethanol). Blood was harvested retro-orbitally using plain micro-hematocrit capillary tubes (Duran Wheaton Kimble) and collected in $K_2EDTA$ capillary tubes (Sarstedt Microvette CB300) 0.5, 1, 2, 4, 8, and 24 h after dosing. Plasma was clarified by centrifugation (4 °C, $450 \times g$, 5 min). To determine tissue concentrations of nirmatrelvir, groups of hamsters were sacrificed 1, 8, and 24 h after dosing and selected tissues were harvested, flash-frozen in liquid nitrogen, and stored at −80 °C. Nirmatrelvir concentrations were analyzed using a qualified LC/MS/MS method, and PK parameters were calculated using non-compartmental analysis with WinNonlin 8.3.3.33. MS analysis defined the analyte area response, internal area response, and transitions (Q1 to Q3). Raw MS data are shown in Supplementary Data 2.

### Hamster efficacy studies

Groups of male and female Roborovski dwarf hamsters were anesthetized using ketamine/dexmedetomidine and infected with $10^4$ pfu per animal (50 µl total volume, 25 µl per nare). Anesthesia was reversed with atipamezole. Hamsters were monitored daily (body weight, temperature, and clinical score). Treatment started 12 h after infection with nirmatrelvir (200 µl, 0.5% MC with 2% Tween80; 250 mg/kg or 125 mg/kg b.i.d.) and ritonavir (50 µl, 20% EtOH; 83.3 mg/kg, b.i.d.). Treatment continued twice daily for 5 and 7 days after infection for VOC δ and VOC o, respectively. Treatment was discontinued for VOC δ after 5 days due to treated hamsters developing pronounced diarrhea. Groups of hamsters were euthanized three days after infection to assess lower respiratory tract viral load. Lungs were harvested, washed with sterile phosphate buffered saline, and homogenized in PBS supplemented with antibiotic-antimycotic using a bead blaster (3 bursts of 30 s at 4 °C each, 30 s pause between bursts; Bead Blaster 24R, Benchmark). Lung homogenates were clarified by centrifugation (20,000 × g, 10 min, 4 °C), aliquoted, and stored at −80 °C until processed. For RNA quantification, lungs and tracheas were harvested and homogenized in RNeasy RNA lysis buffer (Qiagen) in accordance with the manufacturers protocol.

## Assessing viral rebound in hamsters

Male and female dwarf hamsters (6 months old) were randomly assigned in different groups. Upon arrival, all animals were rested for 72 h prior to being transferred into single-ventilated cages to an ABSL-3 facility. For infections, animals were anesthetized using ketamine/dexdomitor and infected intranasally with $10^4$ pfu (50 μl) SARS-CoV-2 VOC o (B.1.1.519). Treatment was initiated 12 h post infection using nirmatrelvir (200 μl, 0.5% MC with 2% Tween80; 250 mg/kg) and ritonavir (50 μl, 20% EtOH; 83.3 mg/kg, b.i.d.), molnupiravir (200 μl, 1% MC; 250 mg/kg; b.i.d.) or vehicle (200 μl, 0.5% MC with 2% Tween80). Twice-daily treatment continued until five days post infection. Vehicle animals were euthanized on days 3, 5, and 9, nirmatrelvir and molnupiravir-treated hamsters were euthanized on days 5 and 9. Lungs were harvested, washed with sterile phosphate buffered saline, and homogenized in PBS supplemented with antibiotic-antimycotic using a bead blaster (3 bursts of 30 s at 4 °C each, 30 s pause between bursts; Bead Blaster 24R, Benchmark). Lung homogenates were clarified by centrifugation ($20,000 \times g$, 10 min, 4 °C), aliquoted and stored at −80 °C until processed. Virus titers were determined by $TCID_{50}$ assays.

## Single ascending dose pharmacokinetics in ferrets

Groups of ferrets were administered a single oral dose of nirmatrelvir at a dosage of 20 mg/kg ($n = 3$) or 100 mg/kg ($n = 3$); (2 ml; 0.5% MC with 2% Tween80) with ritonavir (1 ml; 6 mg/kg; 20% EtOH). A dosage of 6 mg/kg ritonavir was selected based on previous studies in ferrets (53). Drug was administered by oral gavage using a 12 french catheter (Cure Medical). After each gavage, the catheter line was flushed using high calorie liquid diet (3.5 ml; DYNE, PetAg). Blood was harvested in $K_2$EDTA capillary tubes (Sarstedt Microvette CB300) at 0.25, 0.5, 1, 2, 4, 6, 8, and 12 h after dosing. Plasma was clarified by centrifugation (4 °C, $450 \times g$, 5 min), aliquoted and stored at −80 °C. To determine tissue concentrations of nirmatrelvir, groups of ferrets were sacrificed 3 and 12 h after dosing and selected tissues were harvested, flash-frozen in liquid nitrogen, and stored at −80 °C. Nirmatrelvir concentrations were analyzed using a qualified LC/MS/MS method, and PK parameters were calculated using non-compartmental analysis with WinNonlin 8.3.3.33. Raw MS data are shown in Supplementary Data 2.

## In vivo efficacy testing in ferrets against VOCs

Groups of ferrets were anesthetized with ketamine/dexmedetomidine and inoculated with $1 \times 10^5$ pfu of SARS-CoV-2 USA-WA1 (lineage A, isolate USA-WA1/2020, BEI cat# NR-52281; 1 ml, 0.5 ml per nare). Following infection, anesthesia was reversed using atipamezole. Treatment was initiated 12 h after infection and continued twice daily until study end (4 days after infection). Ferrets were treated with vehicle ($n = 6$; 0.5% MC with 2% Tween80), nirmatrelvir/ritonavir (20 ($n = 6$) or 100 ($n = 3$) mg/kg nirmatrelvir; 6 mg/kg ritonavir). Treatments were administered by oral gavage using a 12 french catheter (Cure Medical), starting 12 h after infection in consecutive 2 ml (20 or 100 mg/kg nirmatrelvir; 2 ml 0.5% MC with 2% Tween80) and 1 ml (6 mg/kg ritonavir; 1 ml 20% ethanol) dose volumes. Molnupiravir was formulated in 1% methylcelluose and administered in 3 ml dose volume. After each gavage, the catheter line was flushed using high calorie liquid diet (3.5 ml; DYNE, PetAg). Nasal lavages were collected every 12 h after study start in sterile PBS containing antibiotics and antimycotics (Anti-Anti; Gibco). All ferrets were euthanized 4 days after infection and tissues were harvested to determine SARS-CoV-2 titers and the presence of viral RNA.

## Effect of treatment on SARS-CoV-2 contact transmission in ferrets

A group of source ferrets were inoculated intranasally with $1 \times 10^5$ pfu of SARS-CoV-2 USA-WA1 (lineage A, isolate USA-WA1/2020, BEI cat# NR-52281; 1 ml, 0.5 ml per nare). Twelve hours after infections, source ferrets were further divided into groups ($n = 3$), receiving vehicle, molnupiravir (1.25 mg/kg, b.i.d.; 2.5 mg/kg, b.i.d.; or 5 mg/kg, b.i.d.;), or nirmatrelvir (20 or 100 mg/kg, b.i.d. in 2 mL 0.5% MC with 2% Tween80 + ritonavir (6 mg/kg in 1 ml 20% ethanol)), in all cased administered via oral gavage. At 54 h after infection, each source ferret was co-housed with one uninfected and untreated contact ferret. Co-housing continued until 96 h after infection, when sourced ferrets were euthanized and contact ferrets were separated and housed individually. All contact ferrets were monitored for 4 days after separation from source ferrets and then euthanized. Nasal lavages were performed on source ferrets twice daily until co-housing ended. After co-housing started, nasal lavages were performed on all contact ferrets every 24 h. Upon euthanizing ferrets, nasal turbinates were harvested to determine infectious titers and the presence of viral RNA.

## Effect of prophylactic treatment on direct contact transmission

At study onset, one group of source ferrets were inoculated intranasally with $1 \times 10^5$ pfu of SARS-CoV-2 USA-WA1 (lineage A, isolate USA-WA1/2020, BEI cat# NR-52281; 1 ml, 0.5 ml per nare) and isolated from uninfected ferrets. At the same time, twice-daily treatment regimens were initiated on groups of uninfected ferrets using nirmatrelvir (20 mg/kg) + ritonavir ($n = 3$) or nirmatrelvir (100 mg/kg) + ritonavir ($n = 3$), molnupiravir (5 mg/kg; $n = 3$), or vehicle. Nirmatrelvir was administered in 2 ml (20 or 100 mg/kg in 0.5% MC with 2% Tween80) followed by administration of ritonavir in 1 ml (6 mg/kg in 1 ml 20% ethanol). Molnupiravir was administered in 3 ml (1% methylcellulose). The catheter line was then flushed using high calorie liquid diet (3.5 ml; DYNE, PetAg). Treatment was continued for 6 days until study end. Twelve hours after infection, untreated and infected source ferrets were co-housed with treated contact ferrets. Per approved IACUC protocols, maximal density in the ventilated large animal caging system is 3 ferrets/cage. Therefore, an asymmetrical co-housing matrix had to be applied, consisting of a randomly assigned source to contact animal ratio of either 1:1 or 1:2. After co-housing commenced, nasal lavages were collected every 24 h, and ferrets were monitored until study end. Six days after infecting source ferrets, all animals were euthanized, and nasal turbinates were harvested to determine infectious titers and the presence of viral RNA.

## Titration of SARS-CoV-2 in tissue extracts

For virus titration, the organs were weighed and homogenized in PBS supplemented with antibiotic-antimycotic solution (Gibco). Tissues were homogenized using a bead blaster (3 bursts of 30 s at 4 °C each, 30 s pause between bursts; Bead Blaster 24 R, Benchmark). Tissue homogenates were clarified by centrifugation ($20,000 \times g$ for 5 min at 4 °C). The clarified supernatants were harvested, frozen and used in subsequent $TCID_{50}$ or plaque assays. For detection of viral RNA, the harvested organs were homogenized in lysis buffer (Qiagen), and the total RNA was extracted using a RNeasy mini kit (Qiagen). For nasal lavages, RNA was extracted using a Quick-RNA Viral Kit (Zymo) in accordance with the manufacturer's protocols.

## Quantitation of SARS-CoV-2 RNA

SARS-CoV-2 RNA was detected using the nCoV_IP2 primer-probe set (nCoV_IP2-12669Fw ATGAGCTTAGTCCTGTTG; nCoV_IP2-12759Rv CTCCCTTTGTTGTGTTGT; nCoV_IP2-12696bProbe(+) 5′FAM-AGATGTC TTGTGCTGCCGGTA-3′BHQ-1) (National Reference Center for Respiratory Viruses, Institut Pasteur). RT-qPCR reactions were performed using an QuantStudio 3 real-time PCR system using the QuantStudio Design and Analysis package. The nCoV_IP2 primer-probe set was used with Taqman Fast Virus 1-step master mix (Thermo Fisher Scientific) to detect viral RNA. A PCR fragment (nt 12669-14146 of the SARS-CoV-2 genome) generated from viral complementary DNA using the nCoV_IP2 forward primer and the nCoV_IP4 reverse primer (nCoV_IP4-14146Rv CTGGTCAAGGTTAATATAGG) was used to create a standard curve to

quantitate the RNA copy numbers. RNA copy numbers were normalized to the weight of tissues used.

## SARS-CoV-2 genome sequencing

SARS-CoV-2 positive specimens were sequenced using IDT xGen SARS-CoV-2 Amplicon Panel (previously Swift Biosciences SNAP panel), following the manufacturer's protocol. Sequencing reads were processed and analyzed using TAYLOR default settings available at https://github.com/greninger-lab/covid_swift_pipeline (https://doi.org/10.5281/zenodo.6142073). Sequencing reads are available in NCBI BioProject under access code PRJNA894555.

## Statistics and reproducibility

The Microsoft Excel (versions 16.52) and Numbers (version 10.1) software packages were used for data collection. The GraphPad Prism (version 9.4.1) software package was used for data analysis. Reverse transcription RT-qPCR data were collected and analyzed using the QuantStudio Design and Analysis (version 1.5.2; Applied Biosystems) software package. Figures were assembled and generated with Adobe Illustrator (version 27.5). T-tests were used to evaluate statistical significance between experiments with two sets of data. To evaluate statistical significance when more than two groups were compared or datasets contained two independent variables, one- and two-way ANOVAs, respectively, were used with Tukey's, Dunnett's, or Sidak's multiple comparisons post hoc tests as specified in figure legends. Specific statistical tests are specified in the figure legends for individual studies. Supplementary Data 1 summarizes statistical analyses (effect sized, P values, and degrees of freedom) of the respective datasets. Effect sizes between groups were calculated as $\eta^2 = (SS_{effect})/(SS_{total})$ for one-way ANOVA and $\omega^2 = (SS_{effect} - (df_{effect})(MS_{error}))/MS_{error} + SS_{total}$ for two-way ANOVA; $SS_{effect}$, sum of squares for the effect; $SS_{total}$, sum of squares for total; $df_{effect}$, degrees of freedom for the effect; $MS_{error}$, mean square error. Effective concentrations for antiviral potency were calculated from dose-response virus yield reduction assay datasets through four-parameter variable slope regression modeling. A biological repeat refers to measurements taken from distinct samples, and the results obtained for each individual biological repeat are either shown in the figures along with the exact size (n, number) of biologically independent samples, animals, and independent experiments or results are presented as geometric means ± SD along with the exact size (n, number) of biologically independent samples represented. The measure of the center (connecting lines and columns) is specified in the figure legends. The statistical significance level ($\alpha$) was set to <0.05 for all experiments. Exact P values are shown in Supplementary Data 1 and in individual graphs when possible. Quantitative source data for each biologically independent sample are provided in the Source data file.

## Reporting summary

Further information on research design is available in the Nature Portfolio Reporting Summary linked to this article.

## Data availability

The amplicon tiling sequencing reads generated in this study have been deposited in the NCBI BioProject database under accession code PRJNA894555. All other data generated in this study are provided in this article, the supplementary information, supplementary data files, and the Source data file. Source data are provided with this paper.

## Code availability

Sequencing reads were analyzed using the TAYLOR pipeline (available at https://github.com/greninger-lab/covid_swift_pipeline). All commercial computer codes and algorithms used are specified in "Methods".

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

## Acknowledgements

We thank the Georgia State University Department of Animal Resources and High Containment Core for Assistance and J-J Yoon for technical support. This study was supported, in part, by public health service grants AI141222 (to R.K.P.) and AI171403 project 1 (to R.K.P.) and Scientific Core D (to A.L.G.) from the NIH/NIAID.

## Author contributions

R.M.C., C.M.L., and R.K.P. conceived and designed the experiments. R.M.C., C.M.L., J.D.W., A.K., and R.K.P. conducted most of the experiments. R.M.C. created all figure schematics. A.L.G., N.A.P.L., and P.R. performed next-generation sequencing. M.K.A., R.E.K., Z.M.S., and A.A.K. performed mass spectrometry analysis. M.G.N. and G.R.P. provided critical materials. R.M.C., C.M.L., N.A.P.L., A.L.G., and R.K.P. analyzed the data. R.M.C. and R.K.P. wrote the manuscript.

## Competing interests

G.R.P., MGN., and A.A.K. receive licensing fees and royalties based on Emory's sublicense of the molnupiravir technology to Ridgeback Biotherapeutics, and a family member of GRP serves as the Chief Medical

Officer of Ridgeback. This technology is the subject of the research described in this paper. The terms of this arrangement have been reviewed and approved by Emory University in accordance with its conflict-of-interest policies. R.K.P. reports contract testing from Enanta Pharmaceuticals and Atea Pharmaceuticals, and research support from Gilead Sciences, outside of the described work. R.M.C. reports consulting for Merck & Co, outside of the described work. A.L.G. reports contract testing from Abbott, Cepheid, Novavax, Pfizer, Janssen, and Hologic and research support from Gilead Sciences and Merck, outside of the described work. All other authors declare that they have no competing interests to report.
