## [Peer Review File · Nature Communications]

Comparing molnupiravir and nirmatrelvir/ritonavir efficacy and the effects on SARS-CoV-2 transmission in animal modelsREVIEWER COMMENTS

Reviewer #1 (Remarks to the Author):

In this study, to assess the potency of paxlovid and molnupiravir against SARS-CoV-2, the authors examined viral replication and transmissibility after treatment with these drugs in two animal models, i.e., the Roborovski dwarf hamster and the ferret. They observed that both drugs were effective against the primary infection in the Roborovski dwarf hamster model. In the ferret model, although molnupiravir suppressed SARS-CoV-2 transmission, paxlovid had only a moderate impact on virus replication in the upper respiratory tract.

Although the data are generally descriptive, it is important to share this information, which has not been extensively available in these animal models. Some issues need to be addressed, as detailed below.

Major comments

1. The authors need to examine the underlying mechanism as to why paxlovid does not work efficiently in ferret models. Is paxlovid effective in ferret cell lines?
2. Line 78. Please add more rationale for why the ferret model was used. What value does the ferret model have for these studies?
3. It remains controversial to use ferrets as a SARS-CoV-2 transmission model. Therefore, the authors should carefully state the effect of paxlovid against SARS-CoV-2.
4. The authors should describe the limitations of this study.

Minor comments

1. Line 57. Spell out b.i.d.
2. Line 58. Please describe which omicron variant was used.
3. Fig. 1f. Please define the dotted line. Were any samples below the detection limit? If so, a 1-way ANOVA is not appropriate to compare these groups.
4. Line 88. I think Figure 3b should be Figure 2b.

Reviewer #2 (Remarks to the Author):

Cox et al. have examined the in vivo efficacy of nirmatrelvir/ritonavir combinations in the Roborovski dwarf hamster and the ferret models of SARS-CoV-2 infection, presenting both pharmacokinetic data and various analyses of viral replication over time in both models. The authors calculate a human equivalent dose (HED) of nirmatrelvir for both models. They go on to test the HED dose (20mg/kg) and a higher dose (100mg/kg) of nirmatrelvir with a standard co-dose of ritonavir (6 mg/kg) in the ferret model. They show in Figure 2 that both of these doses reduce viral nasal titers in the ferrets. In Figure 3, the HED dose no longer reduces the viral nasal titers of the source ferrets and also does not prevent close contact transmission to untreated contact animals. The high dose nirmatrelvir and single molnupiravir dose selected both reduced viral nasal titers in the source animals and prevented transmission to untreated contact ferrets in this experiment. Finally, the authors show that this HED nirmatrelvir dose, which had no efficacy in Figure 3, does not work as a prophylactic treatment of contact animals when exposed to untreated infected source ferrets, while their selected dose of molnupiravir, which reduced viral nasal titers in Figure 3 by nearly 4 orders of magnitude, did work as a prophylactic in this experiment.

While I understand the authors time and energy in the development of a clinical antiviral therapeutic in molnupiravir, and they should be commended for these efforts, they also need to be fair to nirmatrelvir in their experimental conditions, analysis, and conclusions. The work here is performed well and there are important conclusions in the co-dosing of nirmatrelvir and ritonavir in multiple animal models and the impact of the two most clinically relevant SARS-CoV-2 antivirals on

transmission. But the overall conclusion of the manuscript that "Paxlovid-like nirmatrelvir/ritonavir fails to block SARS-CoV-2 transmission in ferrets", is not supported by the data. The authors will need to make significant changes to their experimental methods and/or conclusions to make them congruent.

1. The comparison of animal models dosages directly to human dosages needs to be performed with great care. These calculations are broadly generalized and it is still unclear how accurate this is on a drug by drug and animal strain by animal strain basis. This is further complicated by differing viral replication kinetics in humans vs animal models, which may require different drug concentrations and kinetics to exert the same antiviral effect. All of this is even further complicated by ritonavir co-dosing. Therefore a pharmacological HED, without taking viral kinetics into account, isn't particularly relevant. For example, Human clinical viral load data suggests a ~ 0.5 log reduction from Molnupiravir and ~ 0.8 log reduction for Paxlovid, but in the experiments performed here in Figure 2 E+F the HED nirmatrelvir reduced nasal turbinate titers by ~ 1 log, while the dose of molnupiravir used in Figure 3 D+E gives at $\sim 3-4$ log reduction in nasal turbinate titers. This comparison does not come close to mimicking clinical conditions.

2. Furthermore, in Figure 3 D+E, the HED Nirmatrelvir group has no impact on nasal turbinate titers or RNA in source animals. This may be due to some variation in formulation and dosing compared to Figure 2 E+F, where a ~ 1 log reduction was observed. Regardless of the cause, it is not surprising that a non-efficacious dose of nirmatrelvir in source animals, would also have no effect on transmission to contact animals. The "high" dose of nirmatrelvir used in Figure 3 still had less of an impact on nasal turbinate titers compared to the selected dose of molnupiravir, ~ 2 log vs $\sim 3-4$ log reduction, respectively. Again, Paxlovid outperforms molnupiravir in the clinic at reducing viral load. Regardless, the "high" dose of nirmatrelvir prevents transmission to contact animals similarly to the molnupiravir dose, 1/3 vs 0/3 transmissions, respectively.

Because viral load is directly correlated with transmission and we cannot accurately compare human dosages to animal model dosages impact on viral titers, these experiments would need to be performed more comprehensively to fairly compare the impact of these drugs on transmission. To compare the two drugs against each other, the experiments would need to be performed at multiple dosage levels, with multiple impacts on viral load. The major comparison would then have to be source viral load reduction to contact animal transmission.

3. The HED dose of nirmatrelvir (20 mg/kg) was also used in Figure 4, looking at prophylactic treatment of contact animals in transmission studies. In Figure 3, this HED nirmatrelvir dosage has no impact on viral titers, therefore I am not surprised that it was also not effective as a prophylactic treatment. This prophylactic protection experiment would need to be performed with an efficacious dose of nirmatrelvir to make the comparison. Furthermore, some infected animals should be included (separately housed), and nasal turbinate titers quantified, in the contact groups to ensure that the prophylactic contact groups are receiving an efficacious dose of both drugs.

4. Title needs to be changed to reflect the data.

5. The material and methods section needs more experimental detail.

Reviewer #3 (Remarks to the Author):

Cox et al present a timely and well executed body of work to assess the preclinical efficacy of paxlovid and molnupiravir in two animal models, the dwarf hamster that models a severe clinical outcome from SARS Cov2 infection, and the ferret which is more suitable for studying effects on transmission since infected ferrets display minimal clinical signs.

They take care to perform PK studies in both models to ascertain the doses of active nirmatrelvir achieved are relevant to humans. In fact the dose they use in hamsters exceeds that in humans so the effectiveness of this drug on severe disease may be overestimated here.

Overall this is a useful piece of work and should be shared. However the work largely reports negative

findings, that paxlovid does not interrupt transmission; the other observations about control of clinical signs and that molnupiravir does interrupt transmission are previously reported. Moreover one of the most interesting and to my knowledge unexplained observations about paxlovid treatment is the rebound of virus. The authors do mention anecdotal cases of transmission at rebound in their introduction top of page 4 , but the design of transmission experiment they have used here does not allow that to be assessed since they begin cohousing of sentinels with treated donors (figure 3a) at day 2 before rebound occurs (cf day 4 figure 2c)..

My main request would be to clarify the methods and some of the details in the results by improving the presentations in the figures and making legends more thorough.

Specifically:

1. Figure 1.a please described in the legend what the shaded areas signify? And give the n for animals used in PK studies.
2. In the experiment described in figure 3, it is my understanding from eth methods that treatment began 12 hours after infection. Thus it would be expected that there is no difference in virus shedding for any ferrets at the first data point shown in figure 3b. However, it is very difficult to discern the individual data points in this graph as they are all overlaid, but from what on can see it does appear that already the untreated group has higher infectious titre and the molnupiravir group have the lowest titre. If this were the case then the challenge of virus needed to be controlled by molnupiravir is smaller. I take that there is no significant difference in the values as indicated by the large p values given but bearing in mind that there are only 3 animals per group, this is less convincing. I would prefer to be able to see the values rather than squinting at overlaid circles.
3. I very much appreciate that the authors have bothered to sequence the viruses in nasal lavage of infected ferrets and given some thought to the mustelid associated adaptive mutations they find in S gene (extended figure 3). Once again though I feel too much data is combined into one single graph that then might mask potentially interesting findings. Here I would prefer to see the sequence found in animals from figure 3 separated out in order to understand whether the frequency of ferret adapting mutation in S affects the rate of occurrence of transmission. My understanding is that sentinel animals are exposed on day 2, so the important data point is what is the frequency of adaptive mutations at that time point in each of the source animals and does that relate to whether transmission occurred regardless of drug treatment?
4. I find figure 4a very confusing please try to redraw to clarify. During cohousing are there 3 animals in each cage or 4? If 3 (perhaps for husbandry and licensure purposes) the were the sentinels mixed ? randomly? This is not explained at all in the methods section and although the picture shows 3 per cage all the sentinels appear to have a blue arrow indicating they are drug treated so where do the vehicle treated animals go? Perhaps you only need to relabel the blue triangles to say antiviral or vehicle treatment?
5. Page 3, line 5, please define "EPIC-PEP".
6. Page 4, line 7, in extended data fig. 1, looks like the variants tested were separated to two groups, group 1: WA-1, P.1, B.1.1.6, and B.1.351, and group 2: B.1.617.2, B.1.1.529, BA.4, BA.2.12.1 and BA.2. Are there any difference in sequences between two groups?
7. Page 4, line 9, please define "PK"
8. Page 5, line 62, , the text is inconsistent with fig. 1d, 83% and 50%?
9. Page 5, line 76, is there significant difference between high and low dose nirmatrelvir groups?
10. Page 5 second line I find the use of the work conservative a little out of place here- conservative from which aspect? The dose used in hamsters is 7 times more than that used in humans that seems excessive not conservative.
11. Page 5 line -6? Virus burden in trachea of hamsters is not shown anywhere. Please delete ?
12. Page 5 line -6 3 from page bottom there is also no difference between the 2 drug doses, please acknowledge?
13. Page 6, line 83 Line 83, extra "h"
14. Page 6, Line 88, "fig. 3b" should be "fig. 2b"
15. Page 6, Line 91, the units should be pfu/mL and RNA copies/ul.
16. Page 6, Line 92, "mean" should be "median"
17. Page 6, Line 97, in fig. 2c, d, interesting, the rebound was not observed in Viral RNA copies (2d), do we know why? Was the rebound mainly driven by one animal?
18. Page, Line 114, please describe "detection limit" in the figure legends

Response to Reviewers' Comments

Reviewer #1

Although the data are generally descriptive, it is important to share this information, which has not been extensively available in these animal models. Some issues need to be addressed, as detailed below.

Major comments

1. *The authors need to examine the underlying mechanism as to why paxlovid does not work efficiently in ferret models. Is paxlovid effective in ferret cell lines?*

We agree, and have for this revision performed extensive additional studies to address this question. Our approach to the problem was based on the rationale that, in contrast to the nucleoside analog molnupiravir, the protease inhibitor nirmatrelvir does not require intracellular conversion by host enzymes to achieve bioactivity. We therefore consider host cell species-dependence of the antiviral effect to be highly unlikely. This conclusion is supported by the lack of efficacy of paxlovid in the EPIC-PEP (prophylactic administration against household transmission) clinical trial, which revealed poor efficacy of nirmatrelvir in preventing SARS-CoV-2 invasion of the human upper respiratory tract, whereas anti-SARS-CoV-2 activity of nirmatrelvir in human cells *per se* is undisputed. Accordingly, we proposed in the original manuscript that low nirmatrelvir exposure in upper respiratory tract tissues may be a likely mechanism for poor control of SARS-CoV-2 transmission in our study. For this revision, we have experimentally tested this hypothesis by assessing nirmatrelvir in ferret nasal turbinates and lung at anticipated peak (30 minutes after plasma C_{max}) and trough (12 hours after drug administration) time points. Study results (included in the revised manuscript in the newly added Fig. 3j) demonstrate significantly lower peak nirmatrelvir levels in turbinates as compared to lung tissue, identifying nasal turbinates as a privileged compartment of low nirmatrelvir exposure. Please note that it is not feasible to extend this study to molnupiravir (or any other nucleoside analog), since triphosphates are highly unstable *ex vivo* and do not survive nasal turbinate homogenization. Given effective reduction of virus load in turbinates by molnupiravir (Fig. 3h), however, there was no reason to question NHC-TP exposure in upper respiratory tissues. Since the results of our ferret studies recapitulated the outcome of the EPIC-PEP clinical trial, we discuss in the revised manuscript the possibility that the same mechanism may apply to human patients (page 12, line 267 to page 13, line 272).

2. *Line 78. Please add more rationale for why the ferret model was used. What value does the ferret model have for these studies?*

Our study uses two different animal models of SARS-CoV-2 infection, Roborovski dwarf hamsters and ferrets. The dwarf hamster model recapitulates life threatening severe lower respiratory infection with acute lung injury, whereas the ferret model examines upper respiratory tract infection and viral spread through direct contact transmission. Ferrets are mustelids closely related to mink. Strong support for highly efficient spread of the original SARS-CoV-2 strain among mustelids came from reports early in the pandemic that described large SARS-CoV-2 outbreaks in mink farms with extensive transmission between animals and frequent zoonotic and reverse zoonotic transmission (i.e. PMID: 33172935; PMID: 32553059). No zoonotic transmission of this scale has been described for any rodent species. Recapitulating human disease presentation in younger adults, these studies demonstrated that clinical signs in infected mustelids varied from unremarkable or minor to severe respiratory infection in a small percentage of animals. Characterization of SARS-CoV-2 transmission in ferrets by several groups including ours revealed highly efficient upper respiratory tract infection and rapid viral spread (i.e. PMID: 32259477; PMID: 33273742; PMID: 34424180), establishing the ferret as a drug efficacy model to assess inhibition of SARS-CoV-2 replication in the upper respiratory tract infection and suppression of viral transmission (i.e. PMID: 33273742; PMID: 33597220; PMID: 34741049; PMID: 34855509). We have briefly introduced the rationale for the use of ferrets for transmission studies on page 4, lines 34-36 of the revised manuscript.

3. *It remains controversial to use ferrets as a SARS-CoV-2 transmission model. Therefore, the authors should*

carefully state the effect of paxlovid against SARS-CoV-2.

We are not aware of this controversy. We certainly do not debate that none of the available animal models (mice, Syrian golden hamsters, dwarf hamsters, ferrets, NHPs) recapitulates all stages or manifestations of human disease, necessitating careful selection of a model appropriate for the research question asked. For the reasons outlined above (response to point #2 of this reviewer), we have identified ferrets as a leading model to evaluate pharmacological mitigation of SARS-CoV-2 transmission. The growing body of literature referenced above that covers different drugs in clinical use or at advanced stage of clinical testing supports this view.

In addition to supporting efficient SARS-CoV-2 transmission, ferret pharmacokinetic (PK) properties add to the value of ferrets for drug efficacy studies. The impact of organization and function of the digestive system is often underappreciated when evaluating cross-species performance of orally administered therapeutics. In drug development, dogs are typically used as the highest species in the toxicology package. Ferrets, like dogs, are obligate carnivores, which is reflected by high similarities of their digestive systems, often resulting in similar PK properties, that also often more closely resemble that seen in humans than rodent PK. As a case in point, molnupiravir PK properties are similar in ferrets and dogs, and better recapitulate PK properties in humans than rodent models (PMID: 31645453; PMID: 33649113; PMID: 33561864). Unfortunately, substantially less animal data are available for nirmatrelvir, presumably reflecting that the manufacturer has tightly controlled access to the compound until third party-synthesized material became available recently. Because of the physiological similarities discussed, however, greater predictive power of ferret vs rodent PK for drug performance in humans may apply to paxlovid also.

4. The authors should describe the limitations of this study.

We agree, and have substantially expanded the acknowledgement of study limitations throughout the Discussion section. Specifically, we now consider the uncertainty of drug exposure in human tissue beyond plasma PK performance (page 12, line 270 to page 13, line 271), discuss the kinetic differences in virus replication and disease progression in animal models versus humans (page 13, lines 284-289), emphasize that the dwarf hamster-based results may overestimate drug efficacy against acute lung disease (page 11, lines 221-224), and better explain the drug MOA dependency of the correlation between RNA copies and infectious titers (page 11, line 236 to page 12, line 252).

Minor comments

1. Line 57. Spell out b.i.d.

Abbreviation defined as requested.

2. Line 58. Please describe which omicron variant was used.

Defined. In cell culture studies (extended data figure 1), we tested four lineages (lineage B.1.1.529, lineage BA.2, lineage BA.2.12.1, and lineage BA.4). Of these, VOC omicron lineage B.1.1.529 was used in the dwarf hamster studies.

3. Fig. 1f. Please define the dotted line. Were any samples below the detection limit? If so, a 1-way ANOVA is not appropriate to compare these groups.

Clarified as requested. The problem with the "limit of detection" is that once an experimental signal has reached this threshold, one cannot know whether it stayed at, or dived below, this limit – otherwise, it would not be the "limit of detection". Since we cannot define quantitatively what we cannot measure, we plotted these data points at the limit of detection threshold.

4. Line 88. I think Figure 3b should be Figure 2b.

Thank you. We have corrected the error.

Reviewer #2

[...] They show in Figure 2 that both of these doses reduce viral nasal titers in the ferrets. In Figure 3, the HED dose no longer reduces the viral nasal titers of the source ferrets and also does not prevent close contact transmission to untreated contact animals. [...]

This comment misrepresents the experimental results and is not supported by statistical analysis of the datasets. In both studies (shown in figures 2 and 3, respectively), shed virus titer curves in nasal lavages of animals treated with HED nirmatrelvir + ritonavir follow identical trends, deviating from virus titer curves of vehicle-treated animals approximately 48 hours after the onset of treatment (red vs black curves in figures 2c and 3b, respectively). Seeing the repeated comments by reviewer 2 on this topic that appear to imply different treatment outcomes in the efficacy and efficacy/transmission studies shown in figures 2 and 3, respectively, we have added an additional supplementary figure to the revised manuscript, which directly compares vehicle and HED nirmatrelvir + ritonavir-treated animal groups of both studies. Demonstrated in this newly added supplementary Fig. S4 (copied below for convenience), 2-way ANOVA with Sidak's post-hoc test demonstrated that no statistically significant differences between equivalent treatment groups existed in these studies (supplementary Fig. S4c specifically shows HED paxlovid from both studies, which the reviewer implied had different outcomes):

Supplementary Fig. S4 Cross-study comparison of nirmatrelvir + ritonavir efficacy in ferrets. a-c, Virus titers in ferret nasal lavages of vehicle or nirmatrelvir + ritonavir-treated animals from studies presented in Fig. 2 (solid lines) and Fig. 3 (dashed lines). Shown are an overview of vehicle and all nirmatrelvir + ritonavir dose groups from both studies (a), vehicle-treated animals (b), and HED nirmatrelvir + ritonavir-treated animals (c). Symbols represent, and lines connect, geometric means \pm SD. 2-way ANOVA with Sidak's post-hoc multiple comparison test (b-c); P values are shown.

Remarkably, we noted in both studies that, after an initial reduction, virus titers in the HED nirmatrelvir + ritonavir treatment groups plateaued on study day 3 and/or even showed a slight uptick towards study end (also nicely appreciable in the high dose group (blue solid line) of the study shown in Fig. S4a). Naturally, nasal turbinates were extracted at study end on day 4. The minor study-to-study variations in turbinate virus titers that are referenced by the reviewer reflect this slight movement around highly consistent plateau virus loads in the HED nirmatrelvir + ritonavir-treated animals.

In conclusion, rigorous statistical analysis demonstrates that differences in study outcome do not exist. Rather, our data reveal with remarkable reproducibility for outbred hosts that HED nirmatrelvir + ritonavir is only moderately efficacious in controlling SARS-CoV-2 replication in the upper respiratory tract. Because efficacy is moderate, minor individual variations between animals determined whether titer reduction in HED nirmatrelvir + ritonavir vs vehicle-treated ferrets was statistically significant ($P < 0.05$) or not.

[...] Finally, the authors show that this HED nirmatrelvir dose, which had no efficacy in Figure 3, does not work as a prophylactic treatment of contact animals when exposed to untreated infected source ferrets, while their selected dose of molnupiravir, which reduced viral nasal titers in Figure 3 by nearly 4 orders of magnitude, did work as a prophylactic in this experiment. [...]

It was our intention in the design of the prophylactic treatment study to recapitulate key parameters of the EPIC-PEP trial to better understand reasons for its failure and evaluate the predictive power of the ferret model. Naturally, these questions can only be meaningfully addressed if the nirmatrelvir + ritonavir dose administered yields a drug serum exposure equivalent (or, in our case, slightly higher) to that achievable in humans. Given that mean nirmatrelvir serum exposure in human patients treated with paxlovid is approximately 28,220 hrs×ng/ml, the effect of this exposure level in the animal model (HED in our study returned 32,700 hrs×ng/ml) is clinically relevant and was therefore tested in our study. Following a request of this reviewer, however, we have for this revision also tested the prophylactic effect of high dose nirmatrelvir + ritonavir, which had yielded overall serum exposure of 49,500 hrs×ng/ml. Even under these non-physiological high dose conditions, prophylactic nirmatrelvir + ritonavir did not fully suppress infection. These additional data are shown in the revised and expanded figure 4b-e. We decided against investing even more animals to test increasingly unrealistic higher dose levels of nirmatrelvir + ritonavir because of toxicity concerns and for lack of relevance of this exercise that involves large research animals; whether a hypothetical nirmatrelvir plasma exposure >>50,000 hrs×ng/ml may be protective in a prophylactic setting lacks translational implications when maximal exposure levels safely achievable in humans are below 30,000 hrs×ng/ml.

[...] While I understand the authors time and energy in the development of a clinical antiviral therapeutic in molnupiravir, and they should be commended for these efforts, they also need to be fair to nirmatrelvir in their experimental conditions, analysis, and conclusions. [...]

Drug exposure levels are not fair, but measured in hrs×ng/ml. In clinical pharmacology, comparing two drugs in an animal model based on equal systemic exposure-informed HEDs is considered the physiologically relevant experimental setup. Towards our overarching objective to maximize clinical relevance, we established experimental conditions based on the available clinical data. Equivalent nirmatrelvir and N⁴-hydroxycytidine, the systemic metabolite of molnupiravir, plasma exposure levels in humans and ferrets (see our discussion in the above paragraph) confirmed that we have reached this goal. If anything, our study conditions favored paxlovid, since ferret systemic exposure of nirmatrelvir after HED paxlovid slightly exceeded that achieved in humans, whereas systemic exposure of N⁴-hydroxycytidine was slightly lower in ferrets than humans after HED molnupiravir. We have, however, substantially expanded our previous discussion of other study limitations as detailed above (response to comment #4 of reviewer 1).

The work here is performed well and there are important conclusions in the co-dosing of nirmatrelvir and ritonavir in multiple animal models and the impact of the two most clinically relevant SARS-CoV-2 antivirals on transmission. But the overall conclusion of the manuscript that "Paxlovid-like nirmatrelvir/ritonavir fails to block SARS-CoV-2 transmission in ferrets", is not supported by the data. The authors will need to make significant changes to their experimental methods and/or conclusions to make them congruent.

Thank you for the appreciation of the importance of our work. Specified in the following in detail, we have carried out extensive additional studies guided by the reviewer's recommendation for this revision (new data are presented in the newly added figures 1g-j, 3j, and extended data Fig. 3 and in the substantially expanded 3f-l and 4b-e). In addition, we have modified the title of the manuscript and revised the Results and Discussion sections to accommodate these new data and take the recommendations of all three reviewers into account.

1. The comparison of animal models dosages directly to human dosages needs to be performed with great care. These calculations are broadly generalized and it is still unclear how accurate this is on a drug by drug and animal strain by animal strain basis. This is further complicated by differing viral replication kinetics in humans vs animal models, which may require different drug concentrations and kinetics to exert the same antiviral effect. All of this is even further complicated by ritonavir co-dosing. Therefore a pharmacological HED, without taking viral kinetics into account, isn't particularly relevant. For example, Human clinical viral load data suggests a ~0.5 log reduction from Molnupiravir and ~0.8 log reduction for Paxlovid, but in the experiments performed here in Figure 2 E+F the HED nirmatrelvir reduced nasal turbinate titers by ~1 log, while the dose of molnupiravir used in Figure 3 D+E gives at ~3-4 log reduction in nasal turbinate titers. This comparison does not come close to mimicking clinical conditions.

We could not agree more that care must be taken when evaluating the predictive power of animal models to test drug efficacy. The above is a multi-layer comment that we have dissected into its individual components for our response:

- a) Value of PK-informed HED: interspecies allometric scaling to predict dose-based estimates of equivalent systemic exposure from animal models to humans is indeed one of the most controversial areas in clinical pharmacology (PMID: 27057123). A very basic approach converts human equivalent doses based on body weight alone. A slightly more sophisticated method, that is generally accepted in the field, uses body surface area for normalization. However, all scaling calculations are performed to extrapolate from animal systemic exposure to unknown human exposure in preparation of clinical testing. In contrast, our study did not require any allometric scaling since human plasma exposure levels were known (references 33 and 41 in our manuscript). This constellation is admittedly rare, but establishing equivalent systemic exposure levels in humans and animal model species is, as stated above, the objective of clinical pharmacology in studies using animal models. We consider systemic exposure levels particularly relevant for drugs such as nirmatrelvir that exist in plasma in their bioactive form. We have better explained this strength of our study in the revised manuscript (page 10, line 216 to page 11, line 219).
- b) Complication of ritonavir co-dosing: the need for ritonavir to improve nirmatrelvir PK complicates clinical use of the drug indeed. Since ritonavir's role in paxlovid is reduction of nirmatrelvir metabolism, however, it is in our study already "priced-in" into our calculation of overall nirmatrelvir plasma exposure.
- c) Different viral replication kinetics: we agree with the reviewer that host species-dependent differences in the dynamics of viral host invasion and replication constitute an inherent limitation that applies to all research that employs animal models to study viral infections. We have considered this problem in the revised discussion of possible limitations of our work. We have furthermore discussed that, fortunately, preloading the host with drug, as in the prophylactic dosing study shown in figure 4, greatly mitigates this concern of host species-specific viral replication kinetics after infection (page 13 lines 284-289).
- d) Study results do not come close to mimicking clinical conditions: we would like to reemphasize that indeed great care must be taken when attempting cross-study comparisons. In their critique, the reviewer mixes clinical (virus load based on viral RNA copy numbers) and animal (virus load based on infectious titers) study readouts. Unfortunately, this approach is misleading for a non-chain terminating viral mutagen such as molnupiravir, since only reduction in RNA copies, but not the loss of bioactivity of viral RNA harboring incorporated molnupiravir moieties is considered. The revised figure 3 nicely illustrates this frequently neglected point. In the case of a protease inhibitor such as nirmatrelvir, high-dose nirmatrelvir (100 mg/kg) + ritonavir proportionally reduced both viral RNA copies (Fig. 3d,h) and infectious titers (Fig. 3b,f) by approximately 1.5-2 orders of magnitude. In contrast, ultra low-dose molnupiravir (1.25 mg/kg; newly tested for this revision) had no significant effect on overall viral RNA copies (Fig. 3e,i), but reduced infectious titers by approximately 1.5-2 orders of magnitude (Fig. 3c,g). Our results thus reveal that in the presence of low-dose molnupiravir, synthesis of viral RNA is only marginally affected, but a large proportion of this newly synthesized RNA is bio-inactive. The mechanism of action of molnupiravir furthermore dictates that, regardless of host species, a

calculation of drug efficacy based on RNA copy number reduction alone will consistently underappreciate the true effect of the drug. Based on the correlation between viral RNA copies and infectious particle titers in molnupiravir-experienced samples that we experimentally determined in our study, the “~0.5 log reduction” in RNA copies in clinical trials cited by the reviewer should indeed translate to a several order of magnitude reduction in infectious particle titers after molnupiravir treatment, since in proper comparison (reduction of viral RNA copies to reduction of viral RNA copies) our ferret results closely mimic the clinical data. We have better explained the drug MOA-dependency of the correlation between RNA copies and infectious titers in the revised manuscript. However, we have also emphasized that we do not attempt to claim based on ferret data that molnupiravir acts in a rapidly sterilizing manner also in the human upper respiratory tract (page 13, lines 273-283).

2. Furthermore, in Figure 3 D+E, the HED Nirmatrelvir group has no impact on nasal turbinate titers or RNA in source animals. This may be due to some variation in formulation and dosing compared to Figure 2 E+F, where a ~1 log reduction was observed. Regardless of the cause, it is not surprising that a non-efficacious dose of nirmatrelvir in source animals, would also have no effect on transmission to contact animals. The “high” dose of nirmatrelvir used in Figure 3 still had less of an impact on nasal turbinate titers compared to the selected dose of molnupiravir, ~2 log vs ~3-4 log reduction, respectively. Again, Paxlovid outperforms molnupiravir in the clinic at reducing viral load. Regardless, the “high” dose of nirmatrelvir prevents transmission to contact animals similarly to the molnupiravir dose, 1/3 vs 0/3 transmissions, respectively.

We believe the key points have been discussed in our responses to previous comments of the reviewer. To briefly recapitulate, the effect of HED nirmatrelvir + ritonavir on shed virus titers in Fig. 2c and Fig. 3b is statistically identical (supplementary Fig. S4a-c). The suggested variations in formulations and dose levels are baseless. The minor variation critiqued by the reviewer simply represents the reality of performance of a moderately potent drug in outbred hosts.

We did not arbitrarily define a HED nirmatrelvir + ritonavir dose, but based our study on the available clinical data of systemic drug exposure in humans, which is the gold-standard in pharmacology. The HED dose we determined was the dose that resulted in slightly higher overall nirmatrelvir plasma exposure in ferrets than is achievable in humans. We do not know what could be more relevant for predicting the impact of treatment.

Comparisons of clinical performance are based on reduction of viral RNA copies and not infectious titers, which underappreciates efficacy of molnupiravir. Please compare RNA copies (Fig. 3d,e) – they recapitulate the clinical results; then compare the corresponding infectious titers (Fig. 3b-c), which are the relevant parameter for transmission – they reveal the reason why at physiologically relevant dose levels HED molnupiravir, low dose molnupiravir, and even ultra low-dose molnupiravir suppresses SARS-CoV-2 transmission in ferrets, whereas HED nirmatrelvir + ritonavir does not.

Because viral load is directly correlated with transmission and we cannot accurately compare human dosages to animal model dosages impact on viral titers, these experiments would need to be performed more comprehensively to fairly compare the impact of these drugs on transmission. To compare the two drugs against each other, the experiments would need to be performed at multiple dosage levels, with multiple impacts on viral load. The major comparison would then have to be source viral load reduction to contact animal transmission.

This is an excellent suggestion that we have experimentally tested for this revision. The new results are included in the revised Fig. 3 (revised panels Fig. 3c,e,g,i). To briefly summarize, treatment with molnupiravir at ultra low-dose 1.25 mg/kg b.i.d. – 4-fold below HED – fully blocked SARS-CoV-2 transmission, whereas partial transmission still occurred when source animals were treated with high-dose nirmatrelvir + ritonavir at 100 mg/kg b.i.d. (5 × HED). Our data show that not only static viral load at the initiation of co-housing determines transmission success, but also the dynamic changes in virus load during the co-housing period. Bottom line, sustained shed viral titers of $>10^3$ TCID₅₀ units/ml lavage resulted in transmission to at least some contacts, sustained shed viral titers of $>10^4$ TCID₅₀ units/ml lavage resulted in transmission in all study pairs.

3. The HED dose of nirmatrelvir (20 mg/kg) was also used in Figure 4, looking at prophylactic treatment of contact animals in transmission studies. In Figure 3, this HED nirmatrelvir dosage has no impact on viral titers, therefore I am not surprised that it was also not effective as a prophylactic treatment. This prophylactic protection experiment would need to be performed with an efficacious dose of nirmatrelvir to make the comparison. Furthermore, some infected animals should be included (separately housed), and nasal turbinate titers quantified, in the contact groups to ensure that the prophylactic contact groups are receiving an efficacious dose of both drugs.

Again, we did not arbitrarily define a HED nirmatrelvir + ritonavir dose, but used a clinically-informed dose that yielded a slightly higher overall nirmatrelvir systemic exposure in ferrets than is achieved in humans. Although not relevant for clinical reality, we have carried out an additional prophylactic treatment study for this revision, this time dosing the contacts with high-dose nirmatrelvir + ritonavir (100 mg/kg b.i.d., equivalent to $5 \times$ HED). Even under those non-physiologically extreme dosing conditions, one of the treated contacts harbored some infectious viral particles in nasal turbinates at study end. We have included these results in the revised Fig. 4b-e and updated the text accordingly. As explained in our response to the overall evaluation of this reviewer, we decided against investing even more animals to test increasingly unrealistic higher dose levels of nirmatrelvir + ritonavir because of toxicity concerns and for lack of physiological relevance of this exercise that involves large research animals.

4. Title needs to be changed to reflect the data.

We have changed the title to “Human-equivalent dose of paxlovid-like nirmatrelvir/ritonavir does not block SARS-CoV-2 transmission in ferrets”, which accurately captures our key findings.

5. The material and methods section needs more experimental detail.

The Method section in the main text was intentionally reserved for essential study design information in the interest of minimizing the main text word count. A detailed description of study methods had been provided in the Supplementary Information, to which we referred in the main text. For this revision, we have further expanded experimental detail in the Supplementary Information, which we believe is now fully sufficient to reproduce the study if desired.

Reviewer #3

Cox et al present a timely and well executed body of work to assess the preclinical efficacy of paxlovid and molnupiravir in two animal models, the dwarf hamster that models a severe clinical outcome from SARS Cov2 infection, and the ferret which is more suitable for studying effects on transmission since infected ferrets display minimal clinical signs.

They take care to perform PK studies in both models to ascertain the doses of active nirmatrelvir achieved are relevant to humans. In fact the dose they use in hamsters exceeds that in humans so the effectiveness of this drug on severe disease may be overestimated here.

This is a highly valid point. As outlined in our response to comments #2 and 3 of reviewer 1 above, rodent (dwarf hamster) PK differs more substantially from human PK than ferret PK, necessitating substantially higher dose levels. We have added the concern that the dwarf hamster results may overestimate drug efficacy against acute lung disease to the revised discussion of limitations of our work (page 11, lines 221-224).

Overall this is a useful piece of work and should be shared. However the work largely reports negative findings, that paxlovid does not interrupt transmission; the other observations about control of clinical signs and that molnupiravir does interrupt transmission are previously reported. Moreover one of the most interesting and to my knowledge unexplained observations about paxlovid treatment is the rebound of virus. The authors do mention anecdotal cases of transmission at rebound in their introduction top of page 4 , but the design of transmission

experiment they have used here does not allow that to be assessed since they begin cohousing of sentinels with treated donors (figure 3a) at day 2 before rebound occurs (cf day 4 figure 2c).

The duration of SARS-CoV-2 shedding in ferrets is shorter than that in humans and ferrets are poorly susceptible for recent VOC such as delta and omicron, making it difficult to assess rebound of these variants in the ferret model. We have, however, for this revision experimentally explored rebound of VOC omicron replication in the dwarf hamsters after nirmatrelvir + ritonavir or molnupiravir treatment. From peak titers on days 3-4 after infection, virus load rapidly declined in vehicle treated animals. No signs of virus rebound were detected when treatment with nirmatrelvir + ritonavir or molnupiravir was discontinued five days after infection. We have included these additional data in the newly added extended data Fig. 3a-d, described on page 6, lines 97-105.

My main request would be to clarify the methods and some of the details in the results by improving the presentations in the figures and making legends more thorough.

Specifically:

1. Figure 1.a please described in the legend what the shaded areas signify? And give the n for animals used in PK studies.

We have explained in the legend that the reference line represents $10 \times EC_{90}$ in cell culture against different SARS-CoV-2 VOC. This calculation is based on our results shown in the extended data Fig. 1b. Animal numbers in the PK studies are specified in Fig. 1a and Fig. 2a, respectively.

2. In the experiment described in figure 3, it is my understanding from eth methods that treatment began 12 hours after infection. Thus it would be expected that there is no difference in virus shedding for any ferrets at the first data point shown in figure 3b. However, it is very difficult to discern the individual data points in this graph as they are all overlaid, but from what on can see it does appear that already the untreated group has higher infectious titre and the molnupiravir group have the lowest titre. If this were the case then the challenge of virus needed to be controlled by molnupiravir is smaller. I take that there is no significant difference in the values as indicated by the large p values given but bearing in mind that there are only 3 animals per group, this is less convincing. I would prefer to be able to see the values rather than squinting at overlaid circles.

We agree with the reviewer and have revised Fig. 3b-e as follows: we have replaced symbols for each biological repeat (individual animals) with symbols representing geometric group means, and we now show virus titer data for only one lavage per day instead of 12-hour interval data points in the main figure. Since journal policy requires that individual symbols for each biological repeat must be depicted in graphical representations if $n < 10$, we have added a new supplementary figure (supplementary Fig. S3a-b) that shows in enlarged graphs the results of each individual biological repeat for nasal lavages taken in 12-hour intervals. We believe this revised representation complies with journal policy and allows the reader to appreciate that at treatment start (12 hours after infection), viral load was statistically indistinguishable in animals of all groups. The two additional low (2.5 mg/kg) and ultra-low (1.25 mg/kg) molnupiravir groups added for this revision confirm the point.

3. I very much appreciate that the authors have bothered to sequence the viruses in nasal lavage of infected ferrets and given some thought to the mustelid associated adaptive mutations they find in S gene (extended figure 3). Once again though I feel too much data is combined into one single graph that then might mask potentially interesting findings. Here I would prefer to see the sequence found in animals from figure 3 separated out in order to understand whether the frequency of ferret adapting mutation in S affects the rate of occurrence of transmission. My understanding is that sentinel animals are exposed on day 2, so the important data point is what is the frequency of adaptive mutations at that time point in each of the source animals and does that relate to whether transmission occurred regardless of drug treatment?

We have revised the presentation of the genomics results to make these data as accessible as possible (revised extended data Fig. 4a-b). Naturally, all animals were inoculated with virus of the same stock batch. However, there is a higher possibility for species adaptation mutations to appear rapidly in

vehicle-treated animals in which virus replicated unrestricted (high polymerase activity) than in animals in which replication rates are pharmacologically reduced.

4. I find figure 4a very confusing please try to redraw to clarify.

During cohousing are there 3 animals in each cage or 4? If 3 (perhaps for husbandry and licensure purposes) the were the sentinels mixed ? randomly? This is not explained at all in the methods section and although the picture shows 3 per cage all the sentinels appear to have a blue arrow indicating they are drug treated so where do the vehicle treated animals go? Perhaps you only need to relabel the blue triangles to say antiviral or vehicle treatment?

We have revised the study design cartoon (Fig. 4a) for clarification. Our preferred co-housing scheme would have been 4 animals/cage indeed (1 source and 1 each of each contact group (vehicle, paxlovid, and molnupiravir, respectively)). However, ferrets are a USDA-controlled species and housing guidelines are strict. Per our IACUC protocols, maximal approved density in our ventilated large animal caging system is 3 ferrets/cage. We therefore had to apply an asymmetrical co-housing matrix, consisting of a randomly assigned source to contact animal ratio of either 1:1 or 1:2. Since co-housed female ferrets “ball-up” for extended periods of time, different co-housing ratios have in our over 3-year experience with SARS-CoV-2 in this model no effect on transmission efficiency. The revised Fig. 4a schematically shows the different co-housing arrangements.

5. Page 3, line 5, please define “EPIC-PEP”.

Now defined at first appearance; “Evaluation of Protease Inhibition for COVID-19 in Post-Exposure Prophylaxis”.

6. Page 4, line 7, in extended data fig. 1, looks like the variants tested were separated to two groups, group 1: WA-1, P.1, B.1.1.6, and B.1.351, and group 2: B.1.617.2, B.1.1.529, BA.4, BA.2.12.1 and BA.2. Are there any difference in sequences between two groups?

Naturally there are sequence differences, since these lineages have been recognized as variants of concern of the original SARS-CoV-2 lineage. However, none of these lineages carries known resistance mutations to nirmatrelvir in M^{PrO} and all returned active concentrations in the sub-micromolar to nanomolar range, consistent with previous reports.

7. Page 4, line 9, please define “PK”

Defined as requested; ‘pharmacokinetics’

8. Page 5, line 62, the text is inconsistent with fig. 1d, 83% and 50%?

We have corrected the typo; 50% of dwarf hamsters succumbed to VOC omicron infection, not 40%.

9. Page 5, line 76, is there significant difference between high and low dose nirmatrelvir groups?

The difference in dwarf hamster lung virus titers between high and low dose nirmatrelvir treatment groups was not statistically significant (P value 0.7633; 1-way ANOVA with Tukey’s multiple comparison test); now shown in Fig. 1f and stated in the text (page 6, lines 84-87).

10. Page 5 second line I find the use of the work conservative a little out of place here- conservative from which aspect? The dose used in hamsters is 7 times more than that used in humans that seems excessive not conservative.

Agreed, and revised. We meant to express that ritonavir also in the dwarf hamster model succeeds in improving metabolic stability of nirmatrelvir (page 5, lines 61-62).

11. Page 5 line -6? Virus burden in trachea of hamsters is not shown anywhere. Please delete?

We had quantified viral RNA copies in trachea (not enough material to reliably determine infectious titers in dwarf hamster trachea) and lung, and intended to include these data as a supplementary figure, but accidentally missed the figure pointer. For this revision, we have moved the RNA copy results for trachea and lung more prominently to newly added Fig. 1g-j, since they are highly relevant for the viral RNA copies vs infectious particle titer after molnupiravir discussion (see comments of reviewer 2 above).

12. Page 5 line -6 3 from page bottom there is also no difference between the 2 drug doses, please acknowledge?

Acknowledged, and the corresponding P value (see response to comment #9 of this reviewer) was added to the figure.

13. Page 6, line 83 Line 83, extra "h"

Deleted.

14. Page 6, Line 88, "fig. 3b" should be "fig. 2b"

Corrected.

15. Page 6, Line 91, the units should be pfu/mL and RNA copies/ul.

Absolutely, corrected.

16. Page 6, Line 92, "mean" should be "median"

"Mean" is correct, since all virus titers are expressed as geometric means, which is considered to provide a more appropriate representation of log-normally distributed data than the commonly quoted arithmetic mean or the median (<https://www.nature.com/articles/nrg1758>). We have clarified that geometric means are shown in all figures that represent log data.

17. Page 6, Line 97, in fig. 2c, d, interesting, the rebound was not observed in Viral RNA copies (2d), do we know why? Was the rebound mainly driven by one animal?

Actually, we noted an uptick in both virus titer and RNA copies in the high-dose nirmatrelvir + ritonavir group in Fig. 2c-d. When following nasal lavage titers over time in animals of this group individually, each showed an increase in shed virus load towards day 4. The slight shed virus titer uptick in the HED nirmatrelvir + ritonavir group towards study end was likewise driven by several individual animals in this group.

18. Page, Line 114, please describe "detection limit" in the figure legends

Described in the revised figure legends.

REVIEWER COMMENTS

Reviewer #1 (Remarks to the Author):

The authors responded to my review appropriately for the most part; however, one minor issue remains, as detailed below.

Minor comment

For Figures 1f, 3f, and 3h, a non-parametric analysis would be more appropriate for the data that contain unquantifiable data points, as it is difficult to ascertain whether the data have a normal distribution.

Reviewer #2 (Remarks to the Author):

First, I would like to commend the authors for the excellent work performed here and the added experiments that I believe have greatly improved the manuscript. Specifically, the "ultra-low" molnupiravir dosage in the transmission studies have added a more accurate comparison with the "high" Paxlovid-like dose. The added tissue specific PK is also of value. These results are of great importance for publication, but I disagree with the overall conclusions of the authors that a clear difference in transmission between nirmatrelvir and molnupiravir treatment is observed here.

1. I appreciate the original PK work performed here and the additional tissue specific PK added in the revision is of great import to the discussion about the validity of the HED in this transmission study. The use of AUC to predict the HED is a good place to start, but the complications in comparing the efficacy of that HED is fraught with pitfalls:

A. Drug plasma protein binding is not assessed here, if there are significant differences between species (which is observed frequently), this can dramatically alter the amount of free drug available and therefore needs to be incorporated into a HED calculation.

B. Tissue specific penetration can also be a confounding factor across species. It is excellent that the authors have addressed this showing that there is poor nirmatrelvir penetration into the nasal turbinate. Is this poor penetration into the nasal turbinate a ferret-specific problem? Is it a nirmatrelvir-specific problem? We do not know currently, which is an added confounding factor. Is this poor nasal turbinate penetration a major issue for viral transmission in ferrets and humans? I think this would be a very interesting question to answer.

C. The viral replication in ferret cells and therefore the efficacy in nirmatrelvir in ferret cells can certainly be different compared to human cells. These differences are typically not large for a virally-targeted antiviral as the authors point out, but even a 2x difference in EC50 would be important when calculating efficacy in an animal model. If nirmatrelvir is even 50% less effective in ferret cells, the HED loses further relevance. And we don't know the activity in ferret cells.

D. AUC is a great way to compare PK data, but the T1/2 becomes important here as well. Maintaining *free* drug concentrations (again, plasma protein binding is important) above the in vitro EC90 across the entire experiment is critical. The Paxlovid HED used here has a similar AUC to human PK data, but has a higher Cmax and significantly shorter half-life (7 hours for human PK vs 1 hour for ferret PK, Table 1). It looks as if the nirmatrelvir levels drop below the EC90 before the 12-hour mark (Fig 2A) (important for BID dosing). It is hard to tell with the 10x EC90 line (Fig 2A). The line should be changed to a 1x-2x EC90 line and coverage over EC90 in this PK data should be discussed in detail by the authors. If nirmatrelvir concentrations drop below the EC90 before the 12 hour mark, I'm not surprised the HED dose used here has very limited efficacy. A gap in EC90 coverage gives the virus a chance to rebound.

2. I do appreciate the authors comments on qPCR-based viral loads being deceptive for molnupiravir due to its MoA, this is an important point and clearly shown by the data presented in this manuscript.

There is no direct evidence of how big this effect would be in clinical treatment, but it is unlikely to be "several order of magnitude reduction in infectious particle titers after molnupiravir treatment" suggested by the authors in their rebuttal. This effect would most likely mean molnupiravir's reduction in viral titer would be greater than the reported 0.5 log clinical reduction detected by qPCR. Perhaps it would rise to be similar with the Paxlovid clinical viral load reduction measure by qPCR of 0.8 logs. This hypothesis would align with the similar efficacy of the 2 drugs in the clinic. In the end it is a hypothesis and neither I or the authors know the true reduction in viral titers in human patients, but based on similar patient outcomes, the most likely scenario is that both drugs reduce viral titers at similar rates.

Estimating a HED for animal studies can be very informative. The authors and I do agree that it is a complicated endeavor which must be performed with care. When the viral titer data from your experiments (Fig3b+f vs Fig3c+g) clearly shows that the HED from molnupiravir and the HED from Paxlovid are having drastically different impacts (100-1000x) in your source animals, you need to reconsider whether you have accurately estimated both HEDs. Here I do not think that these HEDs are correct based on the viral titer data, unless Paxlovid is drastically less effective in ferrets. If that is the case, the title of manuscript should be "A HED of Paxlovid has little efficacy in ferrets". I do not feel that labeling these doses as HEDs is correct and that needs to be addressed.

Therefore, because the HEDs are questionable here, a fair comparison of transmission can only occur in source animals that have similar reductions in viral titers from Paxlovid and molnupiravir treatment, mimicking the current clinical conditions. The authors can be commended for providing this data in the revision with the addition of the ultra-low dose of molnupiravir (1.25 mg/kg) and high dose of Paxlovid (100 mg/kg). Both of these treatments have a similar 1-2 log reduction in viral titers in both nasal lavage and turbinate of source animals in Figure 3, similar to the clinical results from both drugs. When the source animals have a similar reduction in viral titers, nirmatrelvir treatment transmitted to 1/3 contact animals and the molnupiravir treatment transmitted to 0/3 contact animals. This difference may be real, molnupiravir may out-perform Paxlovid in preventing viral transmission. But this is not enough data to make sweeping conclusions about Paxlovid's ability to prevent transmission. This is an important topic and will make headlines, the title of the manuscript needs to be changed.

Finally, the authors have strengthened their prophylactic protection study in Figure 4 with the high dose of Paxlovid (100mg/kg), which is protective against transmission in Figure 3 similar to the molnupiravir dose used. I'm not surprised that this "high" dose that reduced viral titers in Figure 3, was also protective as a prophylactic in Figure 4. While the HED of Paxlovid barely impacted titers in Figure 3 and was only partially protective in Figure 4. Again the claim that the HED dose does not protect as a prophylactic is misleading because the HED dose estimate is likely flawed.

The authors work here is very high quality and of great importance, but the strength of the conclusions still need to be tempered and the title must be changed to match the results of a 0/3 compared to 1/3 transmission events. The title of this manuscript will be picked up by media outlets and lead to headlines that will further erode the public's confidence in COVID-19 therapeutics. 1 transmission event is not enough for the claims here.

To recap the key points we can agree on:

1. Transmission is governed by infectious viral titer produced in the source animal.
2. Paxlovid and Molnupiravir have similar efficacy and similar viral RNA reduction in clinical data.
3. Your HED of Molnupiravir reduces viral titers 100x-1000x more than your HED of Paxlovid in source animals (Fig. 3b+f vs 3c+g) in the transmission experiments.

Those 3 things cannot all be true when both HEDs are accurately estimated, unless there is a ferret-specific issue. Therefore, without accurate HEDs, a fair comparison of transmission between Paxlovid and Molnupiravir must have similar reductions in viral titers in source animals.

I would finally like say that the work here is performed well and these transmission studies are

important for the scientific and healthcare communities to understand. If significant changes are made to the title and conclusions, I will support the publication.

Reviewer #3 (Remarks to the Author):

the authors have addressed all my concerns

Response to the Reviewers' Comments

Reviewer #1

The authors responded to my review appropriately for the most part; however, one minor issue remains, as detailed below.

Thank you for this assessment of our first revision.

Minor comment

For Figures 1f, 3f, and 3h, a non-parametric analysis would be more appropriate for the data that contain unquantifiable data points, as it is difficult to ascertain whether the data have a normal distribution.

We agree. For this revision, we have performed nonparametric analysis for figures 1f and 3f-I, which contain groups with unquantifiable (at or below limit of detection) data points. We have updated the P values and figure legends accordingly to reflect these changes.

Reviewer #2

First, I would like to commend the authors for the excellent work performed here and the added experiments that I believe have greatly improved the manuscript. Specifically, the "ultra-low" molnupiravir dosage in the transmission studies have added a more accurate comparison with the "high" Paxlovid-like dose. The added tissue specific PK is also of value.

Thank you for this assessment of the quality and importance of our work. We have added the additional groups to establish a basis for dose level selection guided by the sole common biomarker available across all species including humans, the relative reduction of viral RNA levels in the upper respiratory tract through treatment.

These results are of great importance for publication, but I disagree with the overall conclusions of the authors that a clear difference in transmission between nirmatrelvir and molnupiravir treatment is observed here.

1. I appreciate the original PK work performed here and the additional tissue specific PK added in the revision is of great import to the discussion about the validity of the HED in this transmission study. The use of AUC to predict the HED is a good place to start, but the complications in comparing the efficacy of that HED is fraught with pitfalls:

A. Drug plasma protein binding is not assessed here, if there are significant differences between species (which is observed frequently), this can dramatically alter the amount of free drug available and therefore needs to be incorporated into a HED calculation.

This is a fair criticism. We have considered potential species-specific differences in protein binding as a possible limitation of our study in re-revised manuscript (lines 248-252). However, plasma binding does not affect the conclusion of the tissue exposure data added in the previous revision, which revealed lower exposure of nirmatrelvir in nasal turbinates compared to lung tissue.

We furthermore agree that referring to dose levels resulting in equal systemic exposure as "human-equivalent dose" (HED) was unprecise. Discussed in detail below, we have followed a major recommendation of this reviewer and based dose level selections on equivalent effect size in the ferret model to that seen in human patients. Reflecting this change in data interpretation, we have replaced "HED" with "human effect size-equivalent dose (HESED)" throughout.

B. Tissue specific penetration can also be a confounding factor across species. It is excellent that the authors have addressed this showing that there is poor nirmatrelvir penetration into the nasal turbinate. Is this poor penetration into the nasal turbinate a ferret-specific problem? Is it a nirmatrelvir-specific problem? We do not know currently, which is an added confounding factor. Is this poor nasal turbinate penetration a major issue for viral transmission in ferrets and humans? I think this would be a very interesting question to answer.

Having efficacy tested a number of SARS-CoV-2 antiviral candidates in the ferret model, we have not encountered this phenomenon before. However, we agree that lack of precedent does not rule out the possibility of a ferret-specific effect. We have amended the manuscript accordingly, now considering this possibility (lines 313-314). In this context, we have also discussed, however, that clinical trial results suggest that the problem may not be host species- but nirmatrelvir-specific. In the EPIC-PEP study, treatment with paxlovid did not significantly reduce human-to-human transmission (Pfizer EPIC-PEP press release), despite the documented reduced risk of treated patients to advance to severe SARS-CoV-2 lower respiratory disease (Hammond et al. 2022, Arbel et al. 2022). These human data are consistent with the findings presented in our manuscript that paxlovid efficiently mitigates severe COVID-19-like lower respiratory disease (Roborovski

dwarf hamster data; figure 1) but is less impactful in reducing upper respiratory tract infectious virus load (figure 3b).

C. The viral replication in ferret cells and therefore the efficacy in nirmatrelvir in ferret cells can certainly be different compared to human cells. These differences are typically not large for a virally-targeted antiviral as the authors point out, but even a 2x difference in EC50 would be important when calculating efficacy in an animal model. If nirmatrelvir is even 50% less effective in ferret cells, the HED loses further relevance. And we don't know the activity in ferret cells.

We agree that the possibility of host cell-specific effects can never be fully excluded in antiviral drug development. Adding complexity to the problem, there are almost certainly also differences between different ferret cell types and definitely between ferret-derived cell lines, primary ferret cells, and *ex vivo* ferret tissue models. The same applies naturally also to different human cell types and cells derived from any other animal species. Consequently, this issue cannot be fully addressed through any *ex vivo* culture system. Outlined in detail in our responses to the next series of comments of this reviewer, we have, therefore, made treatment effect size equivalent to that seen in human clinical trials the basis for dose level comparisons in the ferret model, following an overarching request of this reviewer.

*D. AUC is a great way to compare PK data, but the T_{1/2} becomes important here as well. Maintaining *free* drug concentrations (again, plasma protein binding is important) above the *in vitro* EC₉₀ across the entire experiment is critical. The Paxlovid HED used here has a similar AUC to human PK data, but has a higher C_{max} and significantly shorter half-life (7 hours for human PK vs 1 hour for ferret PK, Table 1). It looks as if the nirmatrelvir levels drop below the EC₉₀ before the 12-hour mark (Fig 2A) (important for BID dosing). It is hard to tell with the 10x EC₉₀ line (Fig 2A). The line should be changed to a 1x-2x EC₉₀ line and coverage over EC₉₀ in this PK data should be discussed in detail by the authors. If nirmatrelvir concentrations drop below the EC₉₀ before the 12 hour mark, I'm not surprised the HED dose used here has very limited efficacy. A gap in EC₉₀ coverage gives the virus a chance to rebound.*

As stated above, we have replaced “HED” with “human effect size-equivalent dose” (HESED). The concern that T_{1/2} and C_{max} values are not fully identical across ferrets and humans is fair, and has contributed to our decision to base dose level comparisons on equivalent treatment effect size in ferrets and humans as the relevant biomarker. We agree with the reviewer that this approach provides the only viable solution to address the problem of host species-specific differences raised by the reviewer in their comments 1A-1D, including the possibility of differential plasma binding, tissue penetration, viral replication kinetics, and drug half-life. The association between effective drug concentrations in cell culture and acceptable plasma trough *in vivo* is complex and certainly drug-specific and *in vitro* host cell type-dependent. The 10 × EC₉₀ line shown in the original figure 2a was meant to put the PK measurements into some basic perspective for the reader. This objective is equally achieved by showing a 2 × EC₉₀ line. Accordingly, we have revised figures 1a and 2a as requested.

2. I do appreciate the authors comments on qPCR-based viral loads being deceptive for molnupiravir due to its MoA, this is an important point and clearly shown by the data presented in this manuscript. There is no direct evidence of how big this effect would be in clinical treatment, but it is unlikely to be “several order of magnitude reduction in infectious particle titers after molnupiravir treatment” suggested by the authors in their rebuttal. This effect would most likely mean molnupiravir's reduction in viral titer would be greater than the reported 0.5 log clinical reduction detected by qPCR. Perhaps it would rise to be similar with the Paxlovid clinical viral load reduction measure by qPCR of 0.8 logs. This hypothesis would align with the similar efficacy of the 2 drugs in the clinic. In the end it is a hypothesis and neither I or the authors know the true reduction in viral titers in human patients, but based on similar patient outcomes, the most likely scenario is that both drugs reduce viral titers at similar rates.

We fully agree with the reviewer that at present the treatment-mediated reduction of SARS-CoV-2 infectious viral titers in human patients is unknown. Our study demonstrates a discrepancy between reduction of viral RNA copies and infectious viral titers after molnupiravir treatment in two species, one rodent (Roborovski dwarf hamsters) and one non-rodent (ferrets). The lethal viral mutagenesis MOA of molnupiravir is furthermore consistent with a non-linear correlation between viral RNA bioactivity (resulting in viral titer reduction) and physical viral RNA copies (resulting in viral RNA copy number reduction) after incorporation of molnupiravir into the viral genome.

These independent lines of evidence are corroborated by qualitative clinical data available for molnupiravir, which provide at least some insight despite the lack of quantitative viral titer reduction data in humans. Fischer et al have reported that it was impossible to isolate infectious virions from treated patients in a molnupiravir phase 2a clinical trial, whereas infectious virus was successfully recovered from placebo-treated patients (Fischer et al, 2022; PMID 34941423). Although non-quantitative, these clinical data suggest that despite

lowering viral RNA load by only 0.5 log orders, molnupiravir treatment reduced infectious viral particles to unculturable levels in the upper respiratory tract of human patients. We have re-revised the discussion section, now also taking the above evidence and its limitations into account (lines 294-297).

We would like to re-emphasize that we agree with the reviewer that nobody currently knows the true reduction in infectious viral titers in the upper respiratory tract of human patients. The only common quantitative biomarker for treatment effect size available across all species including humans is viral RNA copy number reduction in the upper respiratory tract. Outlined in greater detail in our response to the next series of comments and addressing the overarching request of this reviewer, we have, consequently, in this revision made this biomarker the basis for the selection of comparable dose levels in the ferret model (lines 144-147 and lines 253-256) to experimentally explore two major unknowns, the corresponding reduction in shed infectious viral titers and the effect on viral transmission.

Estimating a HED for animal studies can be very informative. The authors and I do agree that it is a complicated endeavor which must be performed with care. When the viral titer data from your experiments (Fig3b+f vs Fig3c+g) clearly shows that the HED from molnupiravir and the HED from Paxlovid are having drastically different impacts (100-1000x) in your source animals, you need to reconsider whether you have accurately estimated both HEDs. Here I do not think that these HEDs are correct based on the viral titer data, unless Paxlovid is drastically less effective in ferrets. If that is the case, the title of manuscript should be "A HED of Paxlovid has little efficacy in ferrets". I do not feel that labeling these doses as HEDs is correct and that needs to be addressed.

Therefore, because the HEDs are questionable here, a fair comparison of transmission can only occur in source animals that have similar reductions in viral titers from Paxlovid and molnupiravir treatment, mimicking the current clinical conditions. The authors can be commended for providing this data in the revision with the addition of the ultra-low dose of molnupiravir (1.25 mg/kg) and high dose of Paxlovid (100 mg/kg). Both of these treatments have a similar 1-2 log reduction in viral titers in both nasal lavage and turbinate of source animals in Figure 3, similar to the clinical results from both drugs. When the source animals have a similar reduction in viral titers, nirmatrelvir treatment transmitted to 1/3 contact animals and the molnupiravir treatment transmitted to 0/3 contact animals. This difference may be real, molnupiravir may out-perform Paxlovid in preventing viral transmission. But this is not enough data to make sweeping conclusions about Paxlovid's ability to prevent transmission. This is an important topic and will make headlines, the title of the manuscript needs to be changed.

We agree with the reviewer that i) "HED" was an imprecise term that should not have been used, and that ii) basing dose level selections on equivalent antiviral effect size in ferrets to that seen in clinical trials constitutes a superior approach to integrate all available data. We have re-revised the manuscript accordingly.

Specifically, we have:

- Replaced "HED" with human effect size-equivalent dose (HESED).
- Revised the biomarker used for the selection of equivalent dose levels across different host species. In their previous review and this comment, the reviewer requests that the selection of the appropriate ferret dose levels for comparisons should be based on effect size (reduction in virus load) equivalent to that seen in human trials, not PK data. We agree with this rationale. As stated by this reviewer in their original evaluations, clinical trials have shown that treatment effect equates to a reduction of approximately 0.5 log orders of viral RNA copies in the upper respiratory tract through molnupiravir (PMID 34941423), and approximately 0.8 log orders of viral RNA copies through paxlovid (PMID 35172054 and Pfizer press release).

Since viral RNA copy number reduction in the upper respiratory tract is the only quantitative biomarker available across all host species, we queried our experimental data for dose levels corresponding to an equivalent effect size in the ferret upper respiratory tract (lines 9-12, 144-147, and 253-255). Figures 3g and 3d demonstrate that a viral RNA copy number reduction of approximately 0.5 log orders in ferrets was achieved with a dose of 1.25 mg/kg molnupiravir ("ultra-low dose"), and a reduction of 0.8 log order was achieved with 20 mg/kg paxlovid (formerly HED, now renamed "HESED"). Please note that with these dose levels we deliberately erred in favor of paxlovid, since we selected, if in doubt, a higher dose level for paxlovid and a lower dose level for molnupiravir as the appropriate comparison pair.

Having furthermore acknowledged, and discussed, the complexity and limitations of animal models in the re-revised manuscript (lines 251-252 and 325-329), we hope that through these changes we have found common ground with the reviewer of what constitutes a fair, human clinical trial data-driven basis for dose-level comparisons.

We are aware of the potential impact of our study on public perception, and fully agree that sweeping conclusions must be avoided. Following editorial and reviewer guidance, we have changed the title of our manuscript and toned-down major conclusions.

Finally, the authors have strengthened their prophylactic protection study in Figure 4 with the high dose of Paxlovid (100mg/kg), which is protective against transmission in Figure 3 similar to the molnupiravir dose used. I'm not surprised that this "high" dose that reduced viral titers in Figure 3, was also protective as a prophylactic in Figure 4. While the HED of Paxlovid barely impacted titers in Figure 3 and was only partially protective in Figure 4. Again the claim that the HED dose does not protect as a prophylactic is misleading because the HED dose estimate is likely flawed.

As outlined in detail above, we use in the re-revised manuscript equivalent effect size in the ferret model to that seen in clinical trials as the basis for the selection of comparable ferret dose levels and have identified viral RNA copy number reduction in the upper respiratory tract as the sole common quantitative biomarker available across all species including humans. We have made these major changes because the reviewer has made a compelling argument that equivalent effect size constitutes the best parameter for a meaningful, apples-to-apples, comparison. As specified in our response to the previous point of this reviewer, the dose levels identified in figure 3 that satisfy this request in the ferret model are 1.25 mg/kg molnupiravir ("ultra-low dose") and 20 mg/kg paxlovid ("HESED"). Addressing the overarching concern of this reviewer, we have revised the presentation and discussion of the experiments shown in figure 4 accordingly.

The authors work here is very high quality and of great importance, but the strength of the conclusions still need to be tempered and the title must be changed to match the results of a 0/3 compared to 1/3 transmission events. The title of this manuscript will be picked up by media outlets and lead to headlines that will further erode the public's confidence in COVID-19 therapeutics. 1 transmission event is not enough for the claims here.

Again, thank you very much indeed for your in-depth review and the appreciation of the quality and potential impact of our work. It is the overarching objective of our research program to contribute meaningful scientific data to the field, not generating social media buzz or creating headlines. We have tempered our conclusions and, following editorial guidance verbatim, revised the title of our manuscript to "Comparing the effects of molnupiravir and nirmatrelvir/ritonavir on efficacy and SARS-CoV-2 transmission in animal models." Please note, however, that an effect-size based dose level selection using the only quantitative biomarker available across all species including humans has identified 0/3 compared to 3/3 transmission events in ferrets after treatment with molnupiravir or paxlovid, respectively.

To recap the key points we can agree on:

1. Transmission is governed by infectious viral titer produced in the source animal.
2. Paxlovid and Molnupiravir have similar efficacy and similar viral RNA reduction in clinical data.
3. Your HED of Molnupiravir reduces viral titers 100x-1000x more than your HED of Paxlovid in source animals (Fig. 3b+f vs 3c+g) in the transmission experiments.

Those 3 things cannot all be true when both HEDs are accurately estimated, unless there is a ferret-specific issue. Therefore, without accurate HEDs, a fair comparison of transmission between Paxlovid and Molnupiravir must have similar reductions in viral titers in source animals.

This paragraph constitutes a superb summary of our discussion.

- To point 1: We agree. Specifically, shed infectious viral titer in the upper respiratory tract.
- To point 2: We agree with respect to similar viral RNA copy number reduction with slight advantage paxlovid (0.8 log orders for paxlovid versus 0.5 log orders for molnupiravir). We furthermore agree that both drugs have shown similar efficacy to reduce the risk of older patients to progress to severe COVID-19.
- To point 3: Following the reviewer's requests, a dose level selection based on equivalent effect size (reduction of viral RNA copy numbers in the upper respiratory tract) in the ferret model and clinical trials has identified 1.25 mg/kg molnupiravir ("ultra-low dose") and 20 mg/kg paxlovid ("HESED") as the appropriate comparison dose pair in ferrets. This selection is conservative on purpose, since we have, when in doubt, erred on the side of a higher dose for paxlovid and lower dose for molnupiravir. Naturally, effect size relative to vehicle-treated animals shows slight variation at different times after infection. This equivalent effect size-based comparison returned infectious viral titer reductions of 1-2 orders of magnitude for molnupiravir and 0.5-1 orders of magnitude for paxlovid in the upper respiratory tract (figures 3b and 3e). Considering our nirmatrelvir tissue distribution data in ferrets, we anticipate a greater infectious virus titer reduction after paxlovid in the lower respiratory tract, which would be consistent with our Roborovski dwarf hamster results. Although not testable in the ferret model since SARS-CoV-2 does not invade the ferret small airways, this prediction is consistent with clinical efficacy

data that both drugs reduce the risk to advance to severe viral pneumonia approximately equally.

We could not agree more with the reviewer that the exact infectious SARS-CoV-2 titer reduction in human patients treated with paxlovid or molnupiravir is unknown. We hope that we have found common ground that the sole quantitative biomarker available across all host species including humans is the effect of treatment on viral RNA copy numbers in the upper respiratory tract. Following the request of the reviewer verbatim that a fair comparison must be based on a similar effect size in ferrets to that seen in clinical trials, we have in this revision used this biomarker for the selection of appropriate ferret paxlovid and molnupiravir dose levels to experimentally explore the two major unknowns, the corresponding shed infectious viral titers and the effect on viral transmission.

I would finally like say that the work here is performed well and these transmission studies are important for the scientific and healthcare communities to understand. If significant changes are made to the title and conclusions, I will support the publication.

We very much value the reviewer's opinion and are grateful for two insightful reviews. We have changed the title of our manuscript according to editorial recommendations and revised our conclusions guided by the reviewer's argument of what constitutes an appropriate basis for dose level selection. We are optimistic that we can all agree that the sole quantitative biomarker available across all host species is the effect of treatment on viral RNA copy numbers in the upper respiratory tract. We have revised our manuscript based on this rationale (lines 9-12)and re-interpreted our data accordingly.

Reviewer #3

The authors have addressed all my concerns.

Thank you very much indeed for this final evaluation of our work.

REVIEWER COMMENTS

Reviewer #2 (Remarks to the Author):

I appreciate the authors efforts to bring the title more in-line with the results, but the conclusions are still being overstated. The authors agree that predicting a HED is a complex endeavor requiring a comprehensive ADMET data set that is not available here. The authors also agree that the transmission study needs to be benchmarked on upper respiratory viral replication readouts from the source animals. There are two remaining issues that are crucial to the interpretation of the work:

1. The authors have suggested the use of viral RNA measured by qPCR as the benchmark for the transmission studies and this is flawed. It is first flawed by the molnupiravir MoA that we have been discussing. Molnupiravir mutates the viral RNA and therefore does not specifically reduce viral RNA levels. This leads to a discrepancy in the correlation of reductions in viral RNA and infectious titer reductions between the two drugs. Secondly, it has been shown in a similar animal model (Syrian Golden Hamster) that infectious viral titer, and NOT viral RNA levels predicted transmission (<https://doi.org/10.1038/s41586-020-2342-5>). Infectious viral titer should be used to benchmark for these transmission experiments and the conclusion would be that there is no significant difference in transmission, but that the nirm/ritonavir did allow 1 transmission event compared to 0 with molnupiravir at doses that reduced viral titers similarly. If the authors would like to conclude a difference in transmission levels, they will need more data to support that. Again, that 1 transmission event **suggests** a difference in transmission, but is not enough data for a conclusion.

2. The human effect size-equivalent dose (HESED) introduced here has no real description. Is it based on the viral RNA levels? It appears to be still trying to link this to the human dosage when we have agreed that is not the case. These dosages should not be related to the human dose in any way because there is not enough data to make that linkage.

These changes are required for publication, or more data is needed to support the authors conclusion.

Response to the Reviewers' Comments

Reviewer #2

I appreciate the authors efforts to bring the title more in-line with the results, but the conclusions are still being overstated. The authors agree that predicting a HED is a complex endeavor requiring a comprehensive ADMET data set that is not available here. The authors also agree that the transmission study needs to be benchmarked on upper respiratory viral replication readouts from the source animals.

We believe that this discussion has accurately emphasized that the relations between animal models and human data are complex. Following editorial guidance closely, we have for this revision:

- removed comparisons to human dosing from the Abstract, Results, and Figures, and established context to clinical data only in the Discussion section
- referred in the Discussion to viral RNA copy number reduction as a quantitative biomarker to compare the effect of the same drug in the ferret model and the clinic (lines 234-241). Please note that reviewer 1 had explicitly requested to state differences in systemic exposure levels between humans and dwarf hamsters. This statement (lines 260-263) was approved previously by reviewer 1 upon re-evaluation and not removed in this revision.
- resorted to infectious viral titer when assessing the effect of both drugs on viral transmission in the ferret model (lines 276-289)
- avoided definitive conclusions concerning the effect of either drug on human-to-human transmission

There are two remaining issues that are crucial to the interpretation of the work:

*1. The authors have suggested the use of viral RNA measured by qPCR as the benchmark for the transmission studies and this is flawed. It is first flawed by the molnupiravir MoA that we have been discussing. Molnupiravir mutates the viral RNA and therefore does not specifically reduce viral RNA levels. This leads to a discrepancy in the correlation of reductions in viral RNA and infectious titer reductions between the two drugs. Secondly, it has been shown in a similar animal model (Syrian Golden Hamster) that infectious viral titer, and NOT viral RNA levels predicted transmission <https://doi.org/10.1038/s41586-020-2342-5>. Infectious viral titer should be used to benchmark for these transmission experiments and the conclusion would be that there is no significant difference in transmission, but that the nirm/ritonavir did allow 1 transmission event compared to 0 with molnupiravir at doses that reduced viral titers similarly. If the authors would like to conclude a difference in transmission levels, they will need more data to support that. Again, that 1 transmission event **suggests** a difference in transmission, but is not enough data for a conclusion.*

We believe there has been a misunderstanding in this last round of discussion. Nobody has disputed that infectious viral particles carry productive transmission and that shed infectious virus titer in the upper respiratory tract is a correlate for the likelihood of successful transmission. The problem is that infectious virus load in upper respiratory tract of treated human patients is unknown.

In fact, this reviewer has accurately summarized in their second review the major issue that “neither [the reviewer] or the authors know the true reduction in viral titers in human patients” by paxlovid or molnupiravir. In the same review, this reviewer suggested to overcome the problem by assuming that “the most likely scenario is that both drugs reduce viral titers at similar rates”, which is only a guess not backed up by data.

As summarized above, we decided based on editorial guidance to refer in this revision to infectious viral titer when comparing the effect of both drugs on viral transmission in ferrets, and to viral RNA copy number reduction as biomarker when placing the effect in ferrets in context to that of the same drug in the clinic.

2. The human effect size-equivalent dose (HESED) introduced here has no real description. Is it based on the viral RNA levels? It appears to be still trying to link this to the human dosage when we have agreed that is not the case. These dosages should not be related to the human dose in any way because there is not enough data to make that linkage.

The HESED acronym indeed referred to equivalent effect size of viral RNA copy number reduction. We were under the impression that we had found common ground during the course of this discussion that i) effect size is key when placing animal model data in context to clinical results, and ii) that viral RNA copies in the upper respiratory tract is the only quantitative biomarker available for humans and animal models, and therefore the only data-based parameter to establish context of animal and clinical results for the same drug.

That the correlation between viral RNA copy number and infectious viral titer reduction of molnupiravir is non-linear does not invalidate this biomarker, since it directly results from the MOA of molnupiravir independent of the host cell species. In this manuscript alone, we show that the correlation is non-linear in a rodent and non-rodent species, and clinical data that are discussed in the manuscript support, although non-quantitatively, that the correlation is non-linear in human patients also.